*Resource*

# A novel approach to measure mitochondrial respiration in frozen biological samples

Rebeca Acin-Perez[1,2,*,†] (iD), Ilan Y Benador[1,2,3], Anton Petcherski[1,2], Michaela Veliova[1,2,4],
Gloria A Benavides[5], Sylviane Lagarrigue[6], Arianne Caudal[7], Laurent Vergnes[2,8], Anne N Murphy[9] (iD),
Georgios Karamanlidis[10], Rong Tian[7], Karen Reue[2,8], Jonathan Wanagat[2,11], Harold Sacks[1,2],
Francesca Amati[6], Victor M Darley-Usmar[5], Marc Liesa[1,2,4,12] (iD), Ajit S Divakaruni[2,4] (iD),
Linsey Stiles[1,2,†,**] (iD) & Orian S Shirihai[1,2,3,4,12,†,***] (iD)

## Abstract

Respirometry is the gold standard measurement of mitochondrial oxidative function, as it reflects the activity of the electron transport chain complexes working together. However, the requirement for freshly isolated mitochondria hinders the feasibility of respirometry in multi-site clinical studies and retrospective studies. Here, we describe a novel respirometry approach suited for frozen samples by restoring electron transfer components lost during freeze/thaw and correcting for variable permeabilization of mitochondrial membranes. This approach preserves 90–95% of the maximal respiratory capacity in frozen samples and can be applied to isolated mitochondria, permeabilized cells, and tissue homogenates with high sensitivity. We find that primary changes in mitochondrial function, detected in fresh tissue, are preserved in frozen samples years after collection. This approach will enable analysis of the integrated function of mitochondrial Complexes I to IV in one measurement, collected at remote sites or retrospectively in samples residing in tissue biobanks.

**Keywords** frozen tissue; methodology; mitochondrial content; mitochondrial uncoupled respiration; oxygen consumption
**Subject Categories** Membrane & Trafficking; Methods & Resources; Organelles
**The EMBO Journal (2020) 39: e104073**

## Introduction

Mitochondrial oxidative function is an essential parameter to understand metabolism in health and disease. Mitochondria consume 90% of the oxygen that we breathe through Complex IV of the electron transport chain (ETC). The ETC is not only essential to transform the energy of nutrients into a proton gradient used to make ATP, but the ETC is required for all other aspects of mitochondrial-dependent cell metabolism. This is the reason why measuring oxygen consumption, namely respirometry, is the gold standard measurement of mitochondrial function.

A major limitation in oxygen consumption measurements is that they require fresh tissue. There are two key reasons for this: (i) the ETC activity is depressed after freeze–thaw through loss of cytochrome $c$ from the inter-membrane space. (ii) Freeze–thaw damages the mitochondrial membranes, which effectively uncouples the ETC activity (oxygen consumption) from ATP synthesis. These problems are a barrier to basic and translational research since samples cannot be stored and assayed together to decrease the cost and variability of the measurements. This current limitation in oxygen consumption methods restricts measurements from samples stored in biobanks, which are essential for translational research. Consequently, establishing reliable high-throughput methods for assessing mitochondrial function independently of the type of sample and specific freezing methods would overcome this limitation.

1 Department of Medicine, Endocrinology, David Geffen School of Medicine, University of California, Los Angeles, CA, USA
2 Metabolism Theme, David Geffen School of Medicine, University of California, Los Angeles, CA, USA
3 Nutrition and Metabolism, Graduate Medical Sciences, Boston University School of Medicine, Boston, MA, USA
4 Department of Molecular and Medical Pharmacology, University of California, Los Angeles, CA, USA
5 Department of Pathology and Mitochondrial Medicine Laboratory, University of Alabama at Birmingham, Birmingham, AL, USA
6 Department of Biomedical Sciences, University of Lausanne, Lausanne, Switzerland
7 Mitochondria and Metabolism Center, University of Washington, Seattle, WA, USA
8 Department of Human Genetics, David Geffen School of Medicine, University of California, Los Angeles, CA, USA
9 Department of Pharmacology, University of California, San Diego, CA, USA
10 Cardiometabolic Disorders, Amgen Research, Thousand Oaks, CA, USA
11 Department of Medicine, Division of Geriatrics, University of California, Los Angeles, CA, USA
12 Molecular Biology Institute, UCLA, Los Angeles, CA, USA
 *Corresponding author. Tel: +1 310 622 3929; E-mail: racinperez@mednet.ucla.edu
 **Corresponding author. Tel: +1 310 825 8630; E-mail: lstiles@mednet.ucla.edu
 ***Corresponding author. Tel: +1 617 230 8570; E-mail: oshirihai@mednet.ucla.edu
 †These authors contributed equally to this work
[Correction added on 28 May 2020, after first online publication: the author affiliations have been corrected.]

Clinicians have been using spectrophotometric assays to determine the activity of individual ETC complexes or the combination of CI + III or CII + III, in previously frozen samples. These measurements were successfully used in a relatively high-throughput manner to diagnose primary mitochondrial diseases, namely diseases caused by a primary defect in ETC function (Birch-Machin & Turnbull, 2001; Barrientos, 2002; Barrientos *et al*, 2009). However, this approach cannot provide a single measurement of the coordinated function of the ETC function working at more physiological rates. In this regard, some protocols measure supraphysiological activities by using non-physiological electron donors and acceptors. Consequently, spectrophotometric assays might be less sensitive to detect milder reductions in mitochondrial function, such as the ones associated with age-related cardiovascular metabolic diseases. This explains why there have been several attempts to cryopreserve tissues focusing on the maintenance of the mitochondrial inner membrane integrity, by addition of different reagents at the time of freezing with different outcomes (Kuznetsov *et al*, 2003; Nukala *et al*, 2006; Yamaguchi *et al*, 2007; Larsen *et al*, 2012; Garcia-Roche *et al*, 2018). It has been shown that frozen samples do not show coupled respiration as membrane integrity is lost during the freezing process, which results in the uncoupling of electron transport from ATP synthesis. However, despite the loss of coupled respiration, both the enzymatic assays of individual complex activities and the ability of supercomplexes to consume oxygen, after their separation in a blue native gel from frozen samples, support the idea the ETC components are not destroyed by freeze–thawing and the reconstitution of electron transport activity is feasible (Acin-Perez *et al*, 2008). The question arises: Why is it that isolated mitochondria do not respire using the conventional substrate combination, after freezing and thawing if the ETC components are insensitive to freeze and thaw cycles?

We have developed a new approach that reconstitutes maximal mitochondrial respiration in previously frozen samples. Our approach is versatile, as it is amenable to multiple sample types without the need of any special freezing and thawing protocols. In this method, we measure maximal oxygen consumption of the ETC in isolated mitochondria as an integrated unit, using physiological electron donors and acceptors, as in the protocols used for freshly prepared samples. We have also developed a quantitative method to determine mitochondrial content in tissue lysates and intact cells, allowing for evaluation of mitochondrial bioenergetics per cell and also per mitochondria in frozen specimens. Our assay provides a standardized, cost-effective, and widely available test of mitochondrial function, which is one step closer to measure physiological respiration rates in frozen specimens.

## Results

### The electron transport system remains intact in mitochondria isolated from previously frozen liver samples

To determine whether respiratory capacity can be measured in previously frozen liver mitochondria, we first measured their oxygen consumption rate (OCR) using the conventional mitochondrial assay protocols using the Seahorse XF96 Extracellular Flux Analyzer. Mitochondria (4 μg protein/well) isolated from fresh (mFresh) and frozen (mFrozen) mouse livers were assayed in the presence of pyruvate and malate, as substrates, and sequential injections of ADP, oligomycin, FCCP, and antimycin A/rotenone followed. mFrozen assayed using pyruvate and malate as substrates showed significantly lower OCR in all respiratory states compared to mitochondria isolated from fresh tissue (Fig 1A–C). This suggests that when pyruvate plus malate were used as substrates, the electron transfer from Complex I to Complex III and ultimately to cytochrome *c* oxidase is impaired in mFrozen (Fig 1A–C).

We reasoned that freeze–thaw impairs TCA cycle function by disrupting the inner and outer mitochondrial membranes (McGann *et al*, 1988) releasing TCA cycle components from the mitochondrial matrix compartment. While mitochondrial membranes are sensitive to freeze–thaw, published work has shown that the inner membrane mitochondrial supercomplexes maintain assembly and activity after freezing (Acin-Perez *et al*, 2008). We therefore hypothesized that the electron transport system may remain intact in mFrozen. To test this, we assessed respiration in mFrozen using succinate that feeds directly into the electron transport system through Complex II. To control for non-specific respiration, we confirmed that succinate-dependent mFrozen respiration is inhibited by the Complex II-specific inhibitor 3-nitroproprionic acid in a concentration-dependent manner (Appendix Fig S1A). mFrozen pre-incubated with succinate showed a significantly higher respiratory rate than mFresh. However, as compared to mFresh, the mFrozen sample was insensitive to oligomycin. Lack of an oligomycin response in the mFrozen sample is consistent with increased proton permeability as a result of the broken membrane. This is a consequence of the generation of an uncoupled state by the freezing and thawing, leading to lack of control of respiration by Complex V (Fig 1D–F). This conclusion is supported by the observation that mFresh showed increased respiration upon addition of the uncoupler FCCP that reached the same level as mFrozen before the addition of FCCP. Both mFresh and mFrozen were equally sensitive to the Complex III inhibitor antimycin A (Fig 1E and F). These data suggest that electron transport between Complex II, co-enzyme Q, Complex III, cytochrome *c*, and Complex IV is functional in mFrozen.

Next, to test the capacity of other ETC substrates to fuel respiration in mFrozen, we injected the Complex I substrate, NADH. Since intact mitochondria are not permeable to NADH, it is expected that fresh mitochondria will not respire upon injection of exogenous NADH as the sole substrate. In comparison in mFrozen, the inner membrane is permeable and Complex I has direct access to the injected NADH. Indeed, mFresh responded to NADH injection with no significant increase in oxygen consumption (Fig 1G) while pyruvate plus malate, which stimulate the TCA cycle to produce endogenous NADH (Fig 1A–C), resulted in threefold increase in oxygen consumption. In comparison, mFrozen pre-incubated with NADH respired in a similar pattern to that of succinate. In the presence of NADH, respiration of mFrozen was higher than mFresh, insensitive to ATP synthase inhibition and to FCCP, but was sensitive to inhibition by ETC inhibitors (Fig 1G–I). The progressive decrease in NADH-dependent OCR suggests that NADH is rapidly depleted over time in this assay, as we previously observed (Darley-Usmar *et al*, 1987). As a final control, we performed respirometry assays in fresh and frozen liver mitochondria starting in state 1 without any substrates or ADP and injecting substrates (state 4), substrates plus ADP (state 3), and the adenine nucleotide transporter (ANT) inhibitor carboxy-atractyloside

(CAT; Fig 1J and K). As expected, state 1 was similar in fresh and frozen mitochondria and only fresh mitochondria responded to CAT injection as observed with oligomycin. Taken together, our results demonstrate that mitochondria isolated from freeze–thawed mouse livers maintain an intact electron transport capacity despite the disruption of ATP-coupled respiration and substrate shuttle carriers.

## Uncoupled electron transport activity in liver mFrozen is equivalent to mFresh

Our results show that mFrozen are uncoupled but maintain electron transport capacity. However, mFrozen do not respond to the more

commonly used mitochondrial substrates (e.g., pyruvate/malate). We therefore sought to develop a dedicated sequential assay for mFrozen using more compatible substrates. Since mitochondria isolated from frozen tissue are already fully uncoupled, measuring the responses to oligomycin and FCCP is no longer informative (Appendix Fig S1B). Instead, we injected compatible substrates (NADH or succinate) and complex inhibitors (rotenone or antimycin A), respectively. This modification allowed for the inclusion of the substrates needed for Complex IV activity including TMPD/ascorbate and the Complex IV inhibitor, sodium azide. To control for non-mitochondrial oxygen consumption, we confirmed that the TMPD/ascorbate-dependent respiration is inhibited by the Complex IV inhibitor potassium

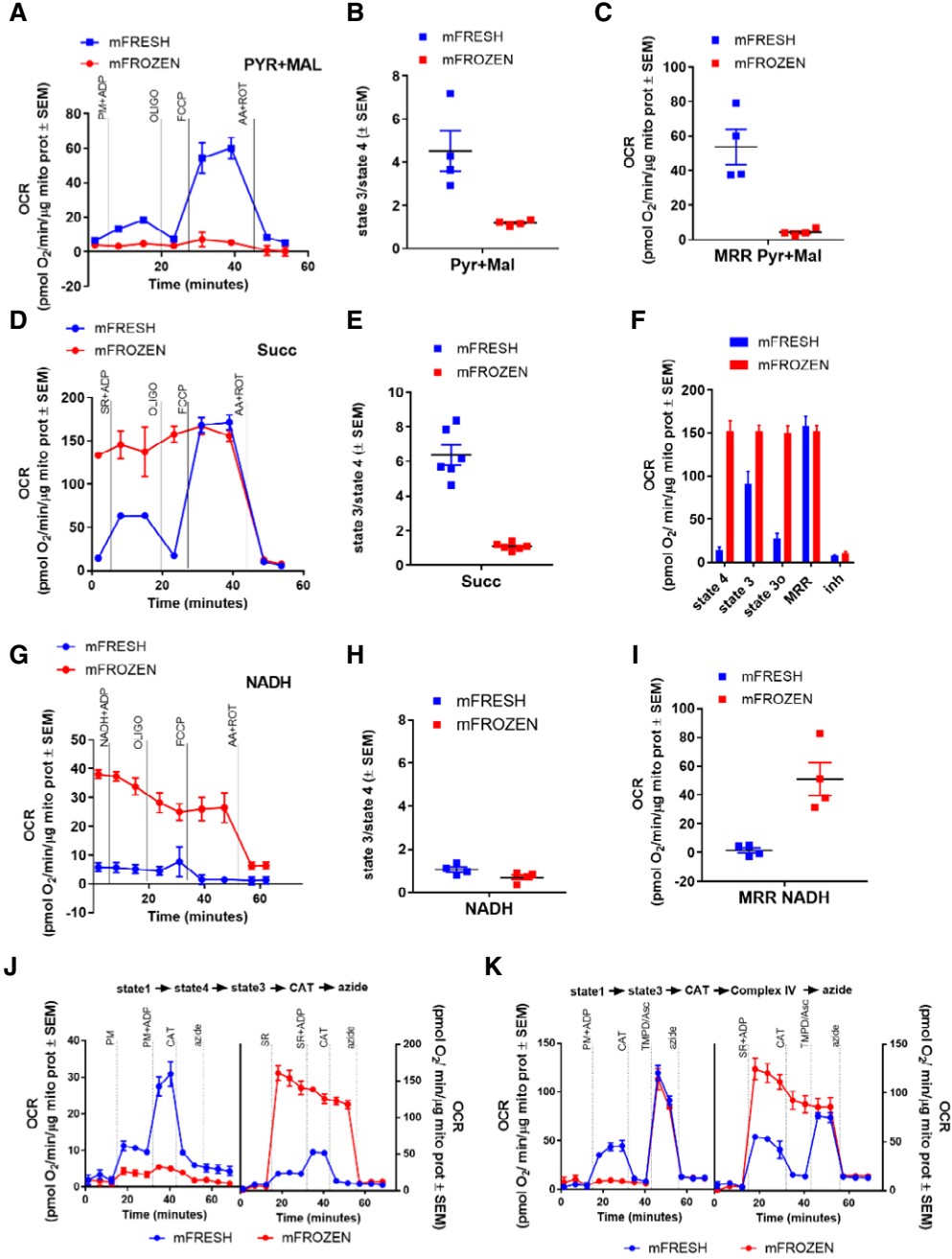

**Figure 1.**

◄

**Figure 1.  Mitochondria isolated from previously frozen liver maintain intact electron transport system.**

A   Representative traces of oxygen consumption rate (OCR) of mouse liver mitochondria isolated from fresh or frozen tissue sustained by pyruvate + malate. Pyruvate + malate + ADP (PM + ADP), oligomycin (oligo), FCCP, and antimycin A + rotenone (AA + ROT) were sequentially injected to assess mitochondrial respiratory states.

B   Pyruvate + malate-dependent state 3 (substrate plus ADP)/state 4 (substrate without ADP) in fresh and frozen liver mitochondria.

C   Quantification of maximal respiration rate (MRR) supported by pyruvate + malate in fresh and frozen liver mitochondria.

D   Representative traces of OCR of liver mitochondria isolated from fresh or frozen tissue supported by the Complex II substrate succinate + rotenone + ADP (SR + ADP).

E   Succinate + rotenone-dependent state 3/state 4 in fresh and frozen liver mitochondria.

F   Quantification of the different bioenergetic parameters sustained by succinate + rotenone in fresh and frozen liver mitochondria.

G   Representative traces of OCR of liver mitochondria isolated from fresh or frozen tissue sustained by the Complex 1 substrate NADH + ADP.

H   NADH-dependent state 3/state 4 in fresh and frozen liver mitochondria.

I   MRR driven by NADH in fresh and frozen liver mitochondria.

J   Representative traces of OCR of liver mitochondria isolated from fresh or frozen tissue starting in state 1 and sustained by substrates without ADP (state 4) and by substrates with ADP (state 3). Mitochondria were tested for CAT sensitivity.

K   Representative traces of OCR of liver mitochondria isolated from fresh or frozen tissue starting in state 1 and sustained by substrates with ADP (state 3). Mitochondria were tested for CAT sensitivity.

Data information: Panels (A, D, G, J, and K) are representative seahorse traces including four technical replicates. Biological replicates: (B and C), $n = 4$; (E and F), $n = 6$; and (H and I), $n = 4$. Every biological replicate represents the average of four technical replicates. Data are the mean $\pm$ SEM.

cyanide in a dose-dependent manner (Appendix Fig S1C). Our results show that mFrozen have a robust azide-sensitive respiratory response to TMPD/ascorbate following injections of any of the three substrates (Fig 2A, C, and E and Appendix Fig S1B).

Since comparison of respiratory chain function in mFrozen and mFresh is constrained by the loss of coupling, we compared activities in the presence of FCCP such that respiration is independent from ATP synthase and so does not require ADP. Our results show that in the presence of pyruvate plus malate, mFresh + FCCP had significantly increased respiration compared to mFrozen, confirming that NADH production by the TCA cycle is impaired in mFrozen (Fig 2A and B). There were no significant differences in TMPD/Asc-dependent respiration between mFresh, mFresh + FCCP, and mFrozen, suggesting that Complex IV maximal capacity of mFresh and mFrozen remained similar. Next, we assessed succinate-dependent respiration, which was essentially identical in mFresh + FCCP and mFrozen, confirming that freeze–thaw uncouples mitochondria but does not damage the mitochondrial electron transport (Fig 2C and D). Finally, we assessed NADH-dependent respiration and found that only mFrozen increased respiration in response to NADH injection (Fig 2E and F). Strikingly, NADH-dependent respiration in mFrozen was quantitatively similar to pyruvate-dependent respiration in mFresh + FCCP (Appendix Fig S1D), suggesting that the assay accurately measured maximal electron transport through intact Complexes I–III–IV. Taken together, these results indicate that we can measure maximal electron transport activity in both Complexes I–III–IV and Complexes II–III–IV as well as isolated Complex IV activity in uncoupled freeze–thawed liver mitochondria. The robust and comparable respiratory response to TMPD/Asc observed under various experimental conditions adds an important internal dimension to this assay allowing for the direct measurement of Complex IV, which facilitates the interpretation of the other complex activities. This overcomes the inherent problems with such comparisons between complexes in the spectrophotometric assays which use different units and kinetic analysis methods to calculate the activity of Complexes I–IV. In addition, the integrated measurement using respirometry also include the contribution of co-enzyme Q to the overall activity. The spectrophotometric assays do have the advantage that they are not potentially constrained by a possible

limiting activity of Complex IV. We propose naming this new respirometry assay protocol as Respirometry In Frozen Samples, or RIFS.

### Optimization of RIFS in tissue homogenates

In the next series of experiments, we optimized the assay for increased compatibility for clinical studies, where sample quantity and dis-synchronized nature of sample collection and access to respirometry equipment in remote sites are limited. Conventional mitochondrial isolation requires a large amount of tissue (from 0.5 to 2 g) and multiple centrifugations in cooled centrifuges depending on the tissues has extremely low yield. In addition, previous studies have observed that isolation of mitochondria from frozen tissue results in the predominant loss of disease-affected mitochondria, masking the pathology. To maximize yield and avoid isolation artifacts (Fernandez-Vizarra *et al*, 2002; Frezza *et al*, 2007; Picard *et al*, 2011; Azimzadeh *et al*, 2016), we sought to determine whether homogenized tissue, cleared of debris by a single centrifugation step, would be compatible with respirometry assays. We assessed the post-nuclear fraction of previously frozen liver homogenate (hFrozen) with pyruvate, succinate, and NADH, using sequential respiratory assay (substrate > oligo > FCCP > AA + ROT). Consistent with results in isolated liver mitochondria, liver hFrozen had lower respiration than hFresh in all respiratory states under pyruvate malate (Fig 3A and B). Respiration dependent on succinate also recapitulated the results in isolated mitochondria: Respiration was higher in hFrozen and was insensitive to oligomycin and FCCP compared to hFresh (Fig 3C and D). Finally, hFrozen had higher respiration on NADH compared to hFresh, consistent with results from isolated mitochondria assay (Fig 3E and F). These results demonstrate that respirometry in the post-nuclear fraction of frozen tissue homogenate is possible, allowing for experimental setups with significantly lower amounts of tissue (as little as 20–50 mg).

In order to assess the comparison in sensitivity between RIFS oxygraphic measurements and spectrophotometric assays, we measured Complex I activity in liver isolated mitochondria and lysates. The spectrophotometric protocol for Complex I measures NADH oxidation in the presence of antimycin A and cyanide, and

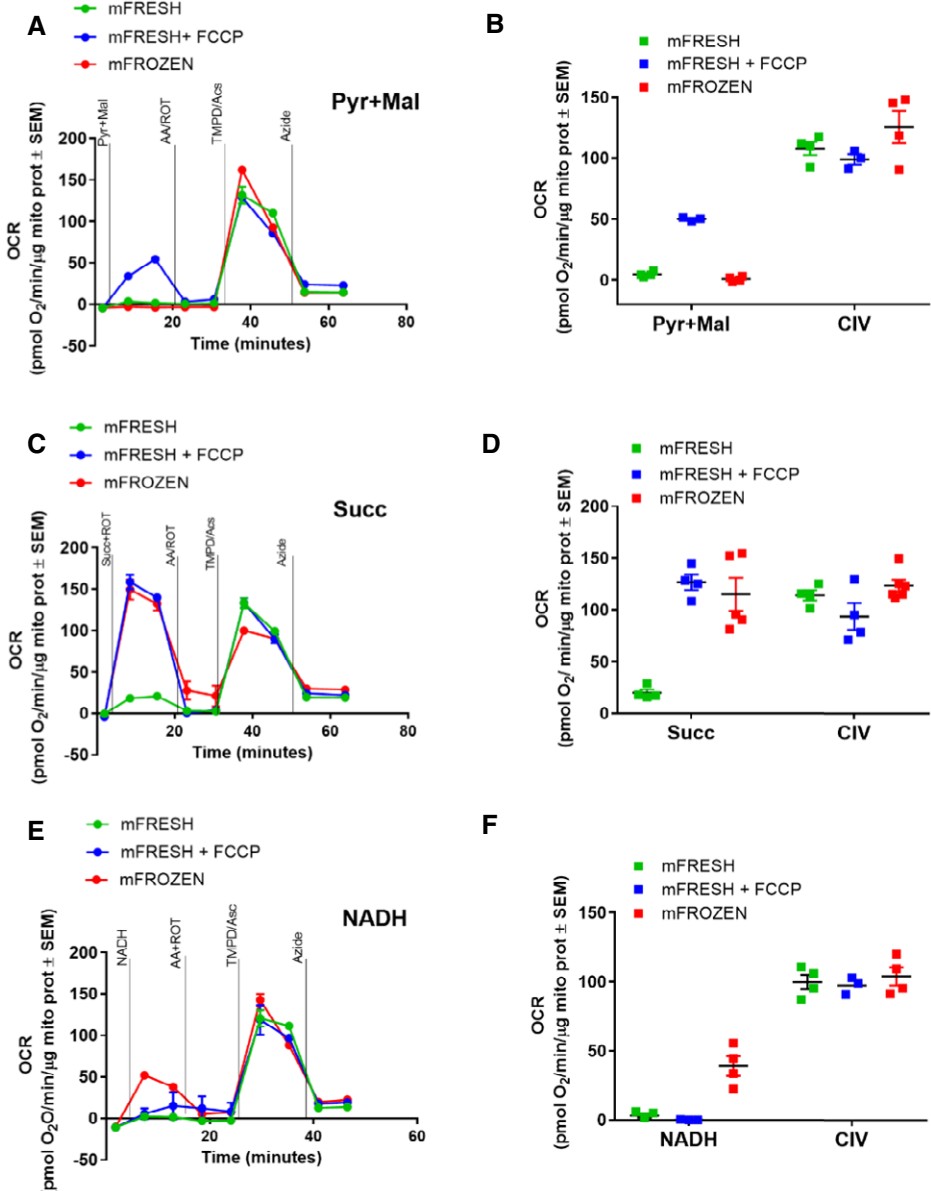

**Figure 2. RIFS measurement of Complex I, II, and IV activity in fresh and frozen liver mitochondria.**

A  Representative pyruvate + malate seahorse profile using RIFS the respirometry protocol in mouse liver mitochondria isolated from fresh or frozen tissue. Pyruvate + malate (Pyr + Mal), antimycin A + rotenone (AA + ROT), TMPD + ascorbate (TMPD/Asc), and azide were injected sequentially

B  Pyruvate + malate- and TMPD/ascorbate (Complex IV, CIV)-dependent respiration in fresh and frozen liver mitochondria.

C  Representative succinate + rotenone seahorse profile using the RIFS respirometry protocol in mouse liver mitochondria isolated from fresh or frozen tissue.

D  Succinate + rotenone (Succ)- and CIV-dependent respiration in fresh and frozen liver mitochondria.

E  Representative NADH seahorse profile using RIFS respirometry protocol in liver mitochondria isolated from fresh or frozen tissue.

F  NADH- and CIV-dependent respiration in fresh and frozen liver mitochondria.

Data information: Panels (A, C, and E) are representative seahorse traces including four technical replicates. Biological replicates: (B and F), *n* = 3–4; (D), *n* = 4–6. Every biological replicate represents the average of four technical replicates. Data are the mean ± SEM.

the rotenone sensitive rate ascribed to Complex I activity. The results of this analysis show that a detectable Complex I activity can be detected in as little as 1 μg of protein in isolated mitochondria and 5 μg in the total homogenate (70–95% rotenone sensitive). In contrast, the spectrophotometric requires a minimum of 100 μg for the homogenate and 20 μg of for the mitochondria with an increase

in error of ± 20% (60% rotenone sensitive). In addition, rotenone sensitivity also depends on the amount of sample (Appendix Fig S2).

Our observation that freeze–thawing permeabilized the inner mitochondrial membrane to ions and metabolites raised the concern whether cytochrome *c*, the only soluble component of the electron transport system, may leak out or be critically diluted in

homogenized preparations. To control for cytochrome *c* concentration, we therefore supplemented samples with exogenous cytochrome *c*. In hFresh, cytochrome *c* supplementation did not

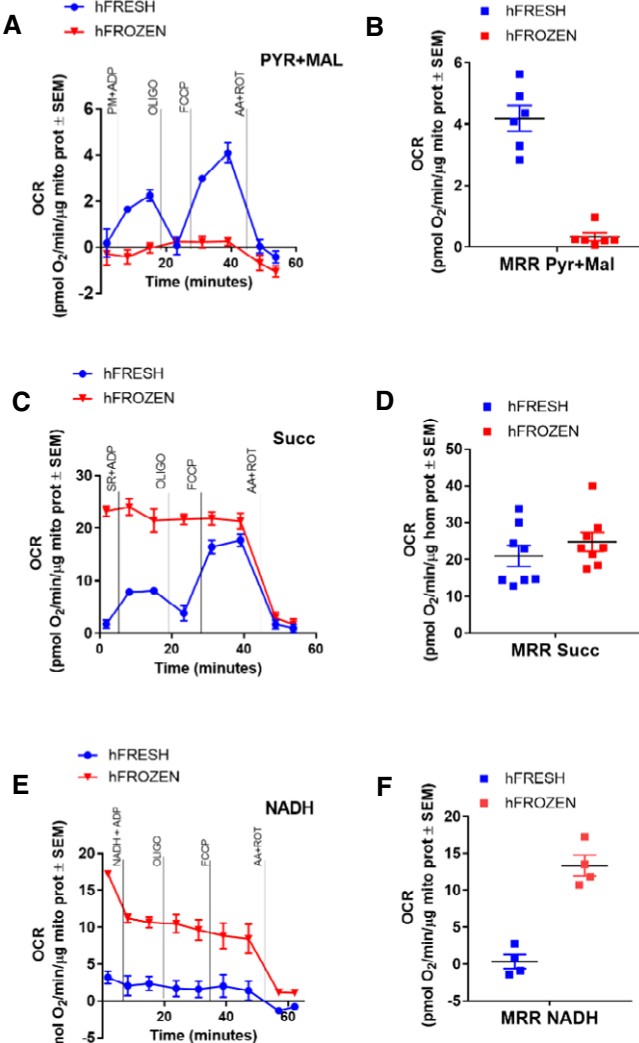

**Figure 3.  Lysates from fresh and frozen tissues maintain intact electron transport system.**

A   Representative pyruvate + malate seahorse profile using the standard respirometry protocol in mouse liver homogenates obtained from fresh or frozen tissue. Pyruvate + malate using the standard respirometry protocol. Pyruvate + malate + ADP (PMA), oligomycin (oligo), FCCP, and antimycin A + rotenone (AA + ROT)

B   MRR dependent by pyruvate + malate in fresh and frozen liver homogenates.

C   Representative succinate + rotenone seahorse profile using the standard respirometry protocol in liver homogenates obtained from fresh or frozen tissue.

D   MRR dependent by succinate + rotenone in fresh and frozen liver homogenate.

E   Representative NADH seahorse profile using the standard respirometry protocol in liver homogenates obtained from fresh or frozen tissue.

F   MRR dependent by NADH in fresh and frozen liver homogenate.

Data information: Panels (A, C, and E) are representative seahorse traces including four technical replicates. Biological replicates: (B), $n = 6$; (D), $n = 8$; and (F), $n = 4$. Every biological replicate represents the average of four technical replicates. Data are the mean $\pm$ SEM.

augment pyruvate- or succinate-dependent OCR, suggesting that cytochrome *c* is not lost during the homogenization process (Appendix Fig S3A and B). In hFrozen, however, cytochrome *c* supplementation significantly increased OCR under all respiratory states, suggesting that freeze–thawing permeabilization causes cytochrome *c* leakage (Appendix Fig S3C and D). For some enzymatic activities, a series of freeze–thawing cycles are performed to ensure that mitochondria are totally broken and accessible to substrates. To test whether several freeze–thawing cycles were needed to perform RIFS, we assessed Complex I activity in the frozen samples with NADH in the presence or absence of digitonin at a concentration that resolves mitochondrial supercomplexes (Appendix Fig S2E). No further increase in NADH-dependent respiration was observed when digitonin was present, supporting the conclusion that the mitochondrial membranes were fully disrupted by one freeze-thaw cycle.

To determine whether cytochrome *c*-supplemented tissue homogenates are compatible with RIFS, we compared hFrozen and hFresh OCR respiration side by side. As observed in isolated mitochondria, when fueled with pyruvate plus malate, hFresh had higher respiration than hFrozen (Fig 4A). When assayed with succinate, hFrozen respiration was higher than hFresh but similar to hFresh + FCCP (Fig 4B). Finally, when exogenous NADH was provided as the sole fuel, respiration was higher in hFrozen compared to hFresh regardless of coupling status (Fig 4C). Interestingly, Complex IV-dependent respiration was consistently higher in hFrozen. We reasoned that Complex IV activity could be decreased in hFresh due to the presence of a residual ATP generating contaminant, resulting in allosteric inhibition of Complex IV. To test this possibility, we measured Complex IV activity in hFrozen in the presence of increasing concentrations of ATP. Our results show that the physiological concentration of 1 mM ATP decreased Complex IV activity to a similar extent as the reduction observed on hFresh (Appendix Fig S3F), suggesting that ATP contamination may explain the discrepancy between Complex IV activity in hFresh and hFrozen. Taken together, these data demonstrate that RIFS analysis can be performed successfully in post-nuclear fractions of tissue homogenates, an approach that markedly simplifies tissue preparation with comparable sensitivity to traditional mitochondrial preparation techniques.

### RIFS can measure physiological changes in respiratory capacity

Our results demonstrate that mitochondrial respiration can be assessed in freeze–thawed samples. To determine whether RIFS can detect pathophysiologically relevant changes, we next assessed previously validated models of pharmacologic and genetic mitochondrial dysfunction. First, we compared the inhibitory effects of metformin and phenformin, two biguanide antidiabetic drugs shown to inhibit Complex I (Bridges *et al*, 2014). As compared to metformin, which is still being prescribed, phenformin was found to have higher cytotoxicity associated with greater potency at Complex I inhibition and induced lactic acidosis (Assan *et al*, 1975; McGuinness & Talbert, 1993). Its clinical use was eventually stopped (Bailey, 2017). If RIFS can detect pathophysiologically relevant respiratory differences, we would expect a significant difference in the respiratory profile of phenformin and metformin. To test this, we performed respirometry in frozen liver mitochondria in the presence of different concentrations of both drugs. Our results show that Ki for Complex I of phenformin is significantly lower than

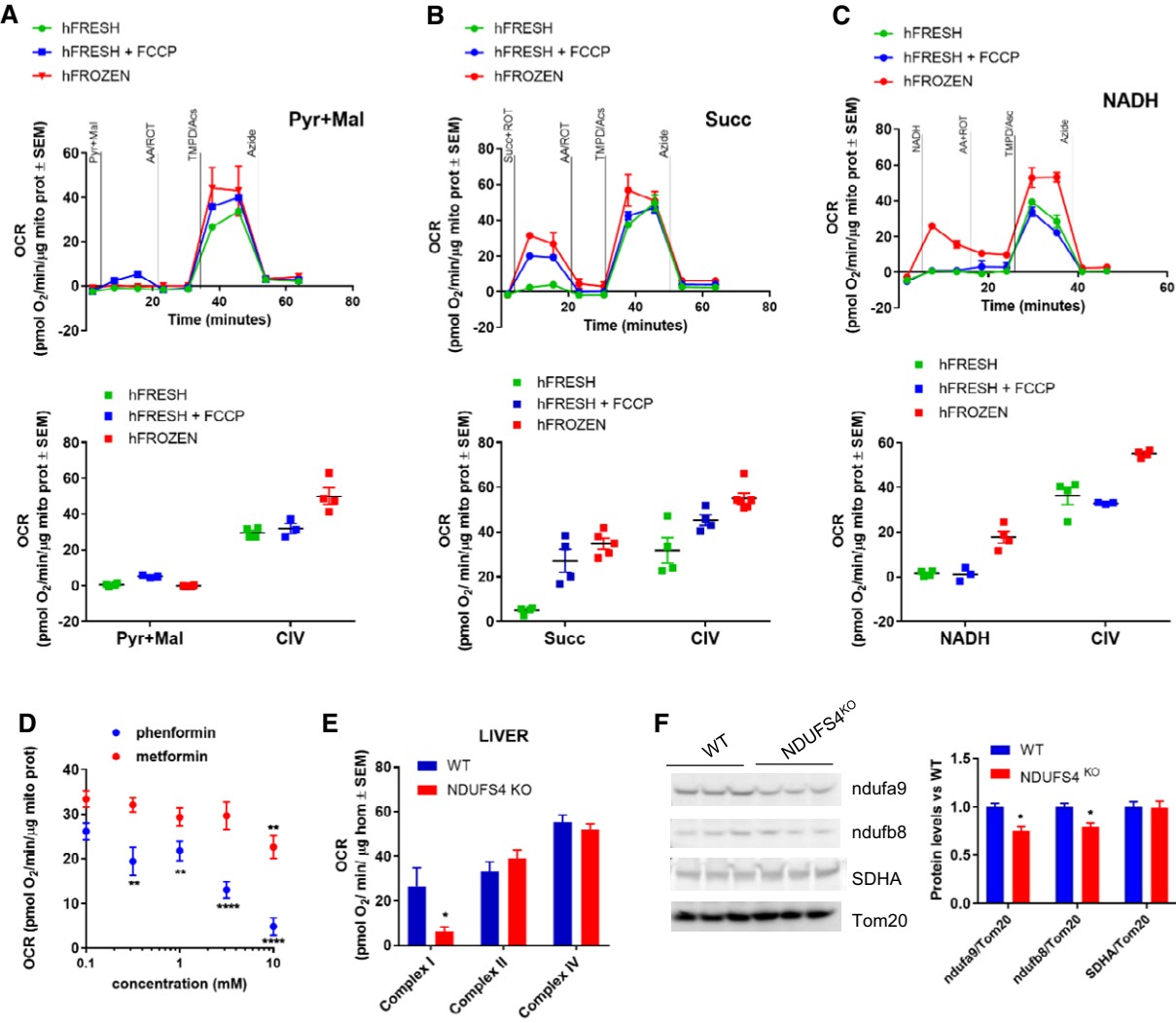

**Figure 4. RIFS measurement of Complex I, II, and IV activity in fresh and frozen liver homogenates.**

A   Representative pyruvate + malate seahorse profile using RIFS respirometry protocol in liver homogenates obtained from fresh or frozen tissue (top panel) and quantification (bottom panel).

B   Representative succinate + rotenone seahorse profile using RIFS respirometry protocol in liver homogenates obtained from fresh or frozen tissue (top panel) and quantification (bottom panel).

C   Representative NADH seahorse profile using RIFS respirometry protocol in liver homogenates obtained from fresh or frozen tissue (top panel) and quantification (bottom panel).

D   NADH-dependent respiration in frozen liver mitochondria in the presence of phenformin and metformin and the indicated concentration.

E   Complex I-, II-, and IV-dependent respiration in liver homogenates from WT and *Ndufs4* KO mice showed impaired Complex I with normal Complex II and IV respiration in mouse *Ndufs4* KO.

F   Representative Western blot followed by quantification of Complex I (NDUFA9 and NDUFB8) levels in *Ndufs4* KO samples. β-actin was used as loading control.

Data information: Biological replicates: (A and C–F), n = 3–4; (B), n = 4–6. Every biological replicate represents the average of four technical replicates. Data are the mean ± SEM. For calculating statistics, the number of replicates used is the biological replicates indicated above. We used the following tests: two-way ANOVA with Tukey's multiple comparison test in panels (D and F); unpaired *t*-test in panel (E). *P < 0.05; **P < 0.01; ****P < 0.0001.

Source data are available online for this figure.

metformin, confirming that RIFS can detect significant pharmacological inhibition of mitochondrial respirometry (Fig 4D).

As an additional validation of RIFS, we next assessed respiratory activity in tissue derived from Complex I-deficient mouse model, *Ndufs4*$^{KO}$ (Karamanlidis *et al*, 2013). Lack of NDUFS4 results in loss of Complex I function due to its poor assembly. Degradation of other Complex I proteins is likely the consequence of failure to assemble

into the complex despite their transcription levels not being affected (Karamanlidis *et al*, 2013). RIFS and Western blot analyses were performed in frozen liver homogenates derived from WT and *Ndufs4*$^{KO}$ mice. Our results demonstrate that Complex I protein levels and Complex I-dependent respiration were specifically decreased in frozen liver mitochondria from mice lacking NDUFS4 while no differences were detected with respect to Complex II nor Complex IV

activities (Fig 4E and F). These results illustrate how RIFS can be used to detect mitochondrial dysfunction secondary to mitochondrial genetic mutations in hFrozen.

### RIFS reveals tissue-specific mitochondrial function

We next optimized RIFS for multiple mouse tissues. First, we performed a parallel comparison between fresh and frozen mitochondria and homogenate in white adipose tissue (WAT). WAT is characterized by low mitochondrial content and therefore requires combining many biological replicates to extract a sufficient sample for any respirometry or bioenergetics assays (Kusminski & Scherer, 2012; Cedikova et al, 2016). This severely limits interpretation of the data as the biologically relevant inter-depot variability and inter-mouse variability are lost. Our results with fresh and frozen WAT mitochondria and homogenates (Appendix Fig S4) are consistent with our results from liver tissue as shown in Figs 2 and 4. As shown in liver, fresh samples have a robust response to pyruvate plus malate and succinate but not NADH, whereas frozen samples responded to NADH and succinate but not pyruvate plus malate.

Next, to improve the utility of RIFS across multiple experimental systems, we assessed RIFS in homogenates from different tissues (Fig 5). Brown adipose tissue (BAT; Fig 5A), heart (Fig 5B), and kidney (Fig 5C) did not require additional optimization and produced robust respiration by following the same protocol as liver and WAT (Fig 5A–C). However, brain and lung, which are very rich in membranes, required additional steps to promote reproducible and consistent respiration data. In these tissues, the standard protocol did not promote total membrane disruption since synaptosomes (Sims & Anderson, 2008) and other membrane rich structures remain intact after homogenization, limiting substrate accessibility to mitochondrial complexes. To promote substrate delivery to the mitochondria, we therefore added alamethicin (ALA) to the assay medium, a channel-forming peptide antibiotic that permeabilizes membranes (Gostimskaya et al, 2003; Matic et al, 2005). In both brain and lung homogenates, the addition of ALA increased respiration dependent on NADH and succinate (Fig 5D and E).

In fibrous tissues such as skeletal muscle, homogenization is more challenging and protocols used for mitochondrial isolation normally require a step of enzyme digestion or mechanical disruption (Kras et al, 2016). Some studies have reported the use of trypsin (Wisniewski et al, 1993; Frezza et al, 2007), collagenase (Rosca et al, 2011), or nagarse (Kras et al, 2016) to help disrupting the muscle fibers prior to homogenization for mitochondrial function assessments and trypsin for individual cardiac fiber isolation (Masson-Pevet et al, 1976; Miersch et al, 2018). To test whether this technique is compatible with frozen samples, we treated soleus muscle with both trypsin and collagenase type II for 10 and 30 min before complete homogenization. Next, we assessed preservation of the integrity of ETC proteins by measuring mitochondrial supercomplex formation in native gels (Schagger & von Jagow, 1991; Acin-Perez et al, 2008; Acin-Perez & Enriquez, 2014) in comparison with the untreated sample (non-enzymatic digestion; Fig 5F and G). While trypsin treatment led to almost complete digestion of supercomplexes (Fig 5F), collagenase treatment preserved supercomplex formation (Fig 5G) without altering mitochondrial content (Fig 5H). Interestingly, collagenase treatment promoted better homogenization

and substrate delivery to mitochondria as reflected in the fourfold increase in mitochondrial respiration using RIFS (Fig 5I).

When we compared frozen homogenate respiration in a battery of different tissues, we demonstrate that RIFS can reveal physiologically relevant differences in OXPHOS capacity (Fig 5J). For example, soleus had significantly higher respiration than quadriceps (SM) as previously described (Jacobs et al, 2013). Furthermore, we confirmed that highly oxidative tissues such as heart, BAT, and brain showed higher mitochondrial respiration rates per milligram of tissue while WAT, known for its low mitochondrial content, showed the lowest rates (Fig 5J). These observations allow us to conclude that RIFS reveals physiological differences consistent with the energy and metabolic demand of the tissue.

### Controlling for mitochondrial mass

Since different tissues contain different amounts of mitochondria per tissue weight, determining mitochondrial mass is essential to interpret respirometry studies in homogenates and permeabilized cells. In addition, in some models of mitochondrial pathology, the cell compensates for the energy deficiency through increased mitochondrial biogenesis (Wredenberg et al, 2002; Sanchis-Gomar et al, 2014; Uittenbogaard & Chiaramello, 2014; Garone & Viscomi, 2018). By including the measurement of mitochondrial mass, we can now refer the mitochondrial activity in the lysates as activity per cell or per mitochondria. While citrate synthase (CS) is commonly employed to assess mitochondrial mass, it is labor-intensive and relies on a single mitochondrial enzyme. Therefore, we sought to develop a high-throughput method to assess mitochondrial content using a mitochondrial dye that would stain mitochondria independently of their membrane potential and their ultrastructure (Cottet-Rousselle et al, 2011). For that purpose, we tested both MitoTracker Red CMXRos (MTR) and MitoTracker Deep Red FM (MTDR) mitochondrial staining in living and fixed cells (Appendix Fig S5). Pancreatic-derived INS-1 cells, were stained with MTR and MTDR and imaged live, after fixation with paraformaldehyde (PFA), and fixation and permeabilization with PFA and Triton X100 (Appendix Fig S5A). Mitochondrial MTDR staining was preserved while MTR diffused after fixation and permeabilization. Similarly, only MTDR showed a competent mitochondrial staining in hepatic Hep2G cells after PFA fixation (Appendix Fig S5B), whereas MTR tended to aggregate outside the mitochondria. In similar fashion, we stained freshly isolated mitochondria with MTR or MTDR before and after depolarizing them with either FCCP or calcium overload and found that membrane potential disruption only affected MTR staining (Fig 6A). Next, we measured MTDR staining in total liver lysate and isolated mitochondria using a standard plate reader. As predicted, MTDR signal per milligram of sample in isolated mitochondria preparation was higher than in tissue homogenates, following a linear correlation before reaching plateau when the signal was saturated at high protein concentrations (Fig 6B and C). Imaging using high-throughput automated microscopy also showed strong correlation of fluorescence intensity and sample amount (Fig 6D), although the plate reader-based assay was more sensitive in revealing differences in the homogenates. We lastly show that MTDR mitochondrial content strongly correlates to mitochondrial protein levels by Western blot in both

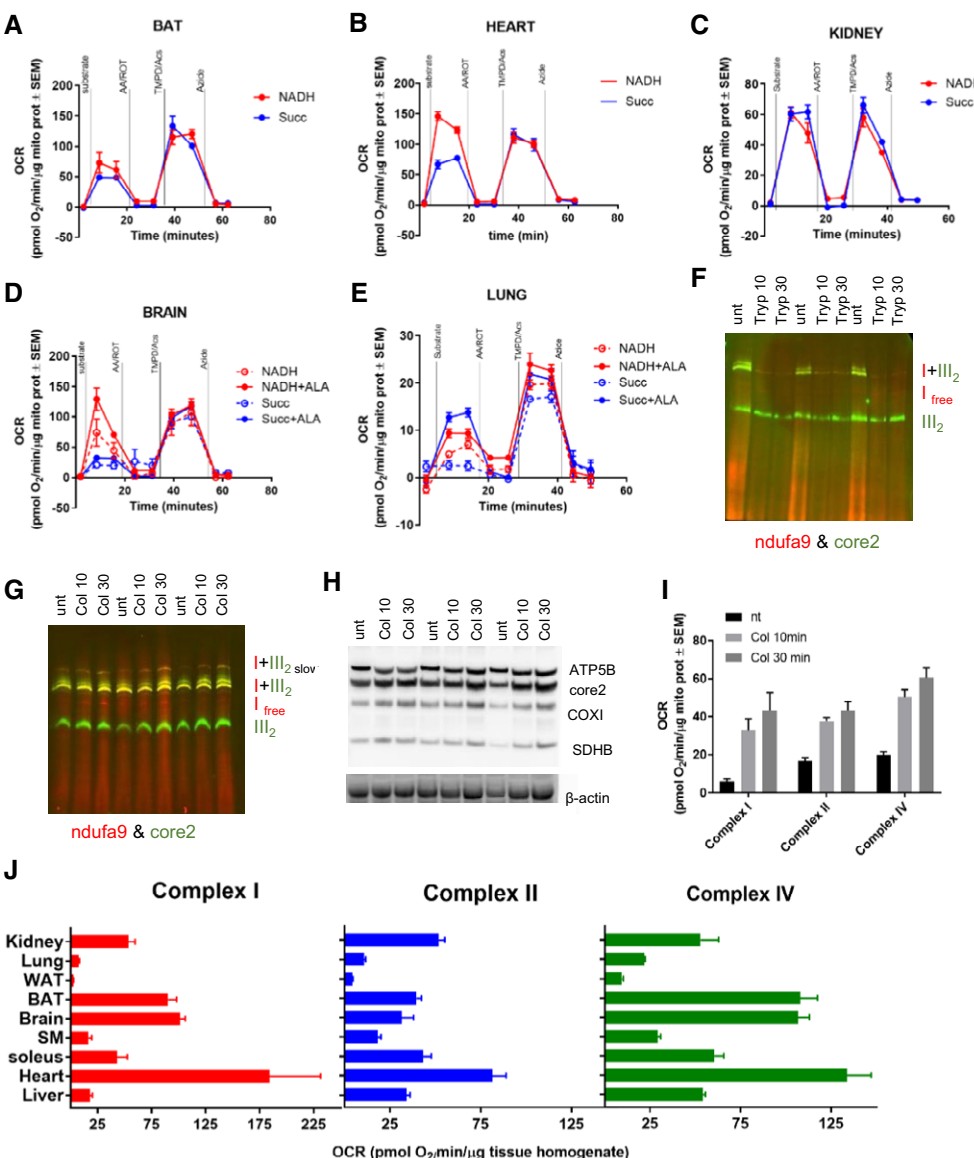

**Figure 5.   RIFS is compatible with a variety of different tissues.**

A–E   Representative seahorse profiles using RIFS respirometry protocol in frozen homogenates from mouse BAT (A), heart (B), kidney (C), brain (D), and lung (E).

F       BNGE and Western blot of soleus muscle incubated with trypsin prior to homogenization.

G      BNGE followed by Western blot of soleus muscle incubated with collagenase type II prior to homogenization.

H      SDS–PAGE followed by Western blot of soleus muscle incubated with collagenase type II prior to homogenization.

I        RIFS respirometry quantification in soleus muscle in the indicated conditions.

J        Complex I-, II-, and IV-dependent respiration in homogenates from frozen tissues from wild-type mice using RIFS respirometry protocol per milligram of protein.

Data information: Biological replicates: (A–E, I and J), $n = 3$–6. Every biological replicate represents the average of four technical replicates. Data are the mean $\pm$ SEM. Western blots illustrate a representative blot of three independent experiments.

Source data are available online for this figure.

mitochondrial, cytosolic, and homogenate samples (Fig 6E and F and Appendix Fig S5C and D). We also measured CS activity as an indicator of mitochondrial mass and compared the activity data to data obtained from MTDR (Appendix Fig S5E). The comparison demonstrates that, at least in the range of 2–30 μg of lysate, the two approaches are linearly correlated.

To test to what extend MTDR could interfere with some remaining blood present in the tissue, we measured MTDR fluorescence in blood

in the presence or absence of MTDR (Appendix Fig S5F and G). Even though there is some noise when high volumes of blood are present, one would not expect that degree of blood contamination when collecting the tissue (tissue is rinsed in PBS prior to freezing).

We next measured mitochondrial content by MTDR fluorescence, CS activity, and Western blot in homogenates from different tissues (Fig 6G–J). Consistent with respirometry data, our results show that that tissues such as BAT and heart showed the highest

mitochondrial content while WAT and lung showed the least (Fig 6G–J). There is not a single method for measuring mitochondrial mass that is encompassing the fact that the various components of mitochondria are expanding equally. One has to select a surrogate marker for mitochondrial mass, and each selection brings with it a potential error that can be addressed in a

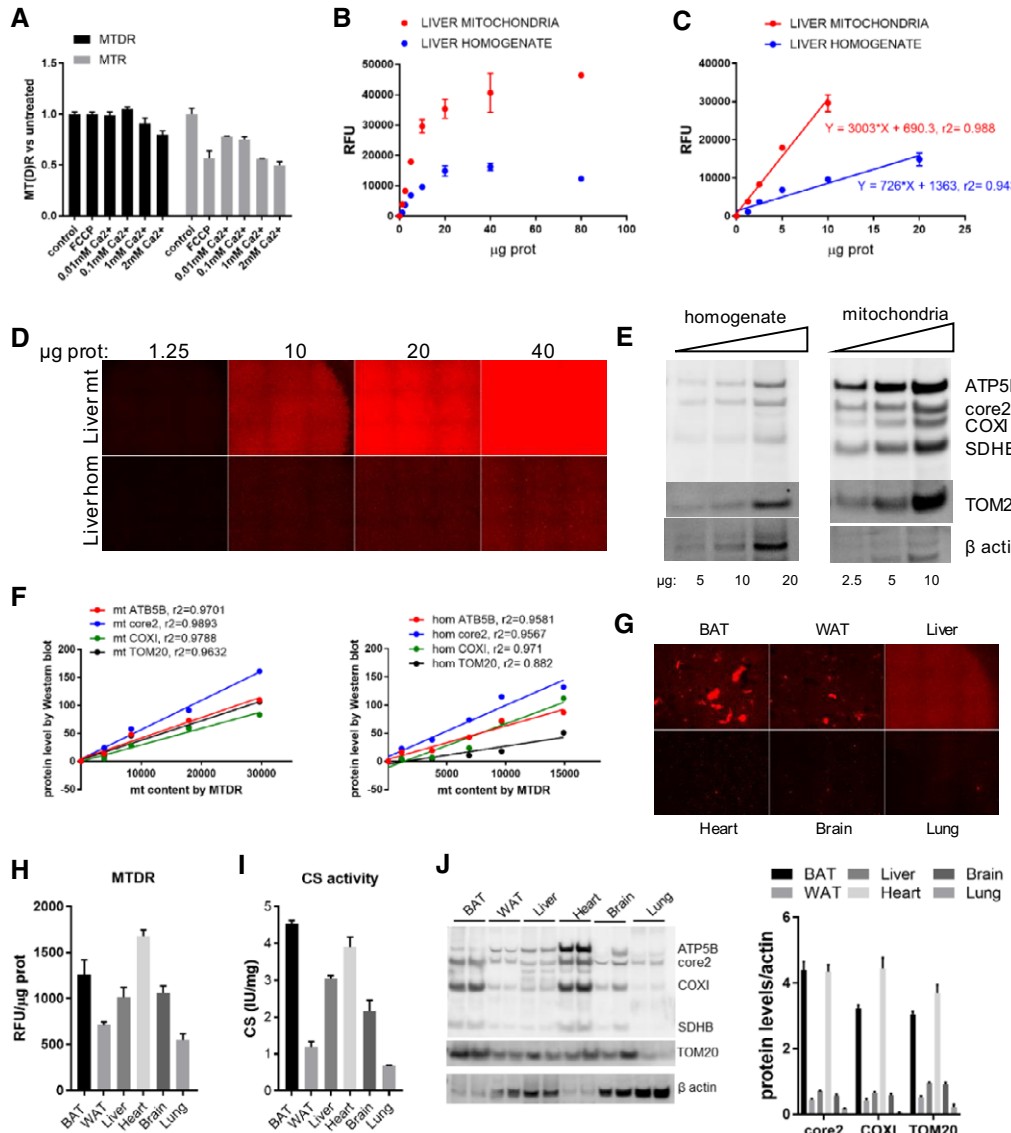

**Figure 6. Mitochondrial content quantification using MTDR.**

A   Fluorescence in freshly isolated mouse liver mitochondria under the indicated treatments and stained with MTDR and MTR. Fluorescence was normalized by protein and referred as percentage of untreated samples.

B   MTDR fluorescence in frozen liver mitochondria and homogenate at the indicated concentration. MTDR fluorescence correlates with protein amount but it saturates at higher protein levels.

C   Linear correlation between MTDR fluorescence and protein abundance in frozen liver mitochondria and homogenate, when the system is not saturated. For this correlation, saturation values have been excluded and only linear range values have been considered.

D   MTDR Operetta imaging in frozen liver mitochondria and homogenate at the indicated concentration.

E   Western blot for mitochondrial protein and actin in increasing concentrations of frozen liver mitochondria and homogenate as indicated.

F   Correlation between MTDR fluorescence (from B and C) and protein levels (from E) in frozen liver mitochondria and homogenate.

G   MTDR Operetta imaging in frozen homogenates of different tissues.

H   MTDR fluorescence in frozen tissue homogenate by microgram of sample.

I   Spectrophotometric citrate synthase activity measure by milligram of lysate.

J   Western blot and quantification of mitochondrial protein by actin (loading control) in tissue homogenates.

Data information: Biological replicates: (A), $n = 4$; (B–F), $n = 5$; and (G–J), $n = 3$. Every biological replicate represents the average of four technical replicates. Data are the mean ± SEM. Western blots illustrate a representative blot of three independent experiments.

Source data are available online for this figure.

thorough analysis of the various components of mitochondria. Among the most commonly used are mtDNA/nDNA ratio, Western blot analysis, and CS activity. However, the one, which is suitable for relatively higher throughput, is a fluorescent-based analysis by either high-throughput imaging or plate reader. We include data with CS activity, WB, and fluorescence analysis of MTDR in a plate reader. The latter approach demonstrates a high-throughput method to assess mitochondrial mass in conjunction to RIFS assay. This is crucial to determine whether reduced respiration detected under RIFS in tissue homogenates is a product of impaired mitochondrial respiration or reduced mitochondrial content.

## RIFS reveals differences in mitochondrial disease models

RIFS optimization in different tissues allows us to study whether there are differences in physiological relevant samples for mitochondrial diseases in other tissues different than liver. $Ndufs4^{KO}$ Complex I-deficient heart homogenates showed specific Complex I impairment (Karamanlidis *et al*, 2013), while Complex II and IV activities were unchanged compared to wild type (Fig 7A). Since mitochondrial content per microgram sample was unchanged (Fig 7B), OXPHOS performance per mitochondria showed the same differences as per homogenate (Fig 7C). Protein steady-state levels of heart lysate further confirmed that Complex I levels were decreased while there was a slight increase in Complex II (Fig 7D), possibly as a metabolic compensation (Acin-Perez *et al*, 2014). We also tested RIFS in heart homogenates in a second mouse model of mitochondrial disease that leads to a progeroid phenotype and cardiomyopathy due to a mutation in the polymerase gamma ($POLg^{MT}$) (Dai *et al*, 2010). Lack of proof-reading activity in $POLg^{MT}$ leads to accumulation of mutations in the mitochondrial DNA (mtDNA)-encoded genes (Trifunovic *et al*, 2004; Kujoth *et al*, 2005). As expected, Complex I was the most affected since mtDNA encodes for 7 proteins of Complex I. Complex II, fully encoded in the nuclear DNA, was slightly increased and Complex IV unchanged in $POLg^{MT}$ heart homogenates (Fig 7E). Since mitochondrial mass was increased in POLgMT (Fig 7F), Complex II and IV activity corrected by mitochondrial content was also decreased (Fig 7G). Protein analysis by Western blot revealed a slight increase in mitochondrial biogenesis (Fig 7H) but not to the level suggested by the mitochondrial determination with MTDR. To assess whether the decrease in mitochondrial function during aging is a general feature, we moved into another experimental animal model, zebra fish. After testing RIFS in zebra fish comparing fresh and frozen muscle homogenate (Appendix Fig S5A), we analyzed mitochondrial activity in young (1 year old) and old (2 years old) zebra fish muscle (Fig 7H). Similar to the phenotype observed in the POLgMT mice, muscle from old zebra fish presented decreased mitochondrial function (Fig 7I). Rotenone treatment has been used acutely in embryos to study the effect on mitochondria in development (Melo *et al*, 2015) as well and in adults to induce Parkinson's disease (Wang *et al*, 2017). We induced mitochondrial dysfunction in dechorionated zebrafish embryos by treating them acutely with rotenone in egg water. Complex I respiration was decreased in a rotenone concentration-dependent manner both using RIFS (Fig 7J and K) and spectrophotometry (Fig 7L). RIFS showed to be more sensitive than enzymatic activity when showing the differences at the lower rotenone concentration. Next, we tested RIFS in human

clinical samples collected intraoperatively from pheochromocytoma patients (Vergnes *et al*, 2016). It has been shown that retroperitoneal periadrenal fat mitochondria from pheochromocytoma patients have higher OXPHOS than controls and express more UCP1, a marker of BAT (Vergnes *et al*, 2016) (Fig 7M). We prepared homogenates from biobanked samples used in the reported study and reproduced the results using RIFS (Fig 7M–P), where pheochromocytoma homogenates showed increased ETC activity, mitochondrial mass, UCP1, and ETC protein levels, as described when browning occurs (Abdullahi & Jeschke, 2016; Cui & Chen, 2016; Winn *et al*, 2019). When intraoperative samples were collected from the hospital, they were placed on ice-cold PBS for 1–3 h prior to freezing. To address how much of the respiratory capacity is preserved/lost when samples are not immediately flash-frozen in liquid nitrogen, we took mouse liver samples and either immediately froze; left it on ice-cold PBS for 0.5 h or 3 h; or left on PBS at room temperature (RT) for 0.5 h or 3 h prior to freezing. We performed RIFS on lysates from the 5 different conditions and found that respirometry was comparable between flash-frozen samples and samples placed on ice-cold PBS before freezing (Appendix Fig S6B). Interestingly, samples placed at RT showed a slight decrease in Complex I activity accompanied by a slight increase in Complex II. This observation together with the results shown in the Western blot, where there is a decrease in Complex I subunits (Appendix Fig S6C), suggests that the respirasome containing Complex I is the first to be degraded at RT. When there is no liquid nitrogen available, the samples should therefore be placed on ice-cold PBS to preserve the integrity and function of the mitochondrial complexes.

The measurement of bioenergetics in cells or platelets isolated from human blood is now emerging as an important aspect of translational mitochondrial biology. In addition, cell culture models have long been a key experimental platform for metabolic research. For these applications, we next optimized RIFS in the human monocytic cell line (THP-1) which was derived from an acute monocytic leukemia patient (Chanput *et al*, 2014; Bosshart & Heinzelmann, 2016). In fresh THP-1 cells, a measurable OCR is evident before the addition of a plasma membrane permeabilizer (PMP) which is not present in the frozen cells (Appendix Fig S6B). This is expected since freeze–thaw will result in the loss of small molecule intermediates from the cell. The addition of PMP in combination with succinate, rotenone, and FCCP preserved Complex II respiration to the same level in both fresh and frozen cells. Next, antimycin A was added to inhibit Complex III, which inhibits succinate-dependent fresh and frozen OCR to the same extent. Complex IV substrates were then added resulting in a stimulation of azide-sensitive OCR to the same level in fresh and frozen cells (Appendix Fig S6D and E). We then tested RIFS in frozen monocytes, lymphocytes, and platelets isolated from healthy donors (Fig 7Q and Appendix Fig S6F–H). In all three cell types, the OCR is low before addition of Complex I, II, or IV substrates and increases substantially in the presence of mitochondrial substrates. In monocytes, lymphocytes, and platelets, the maximal Complex I or II activity also requires the presence of ALA/cytochrome *c*. All three cell types show the anticipated inhibition of succinate-dependent OCR by antimycin A. In contrast, the Complex I-dependent OCR is only partially inhibited by rotenone consistent with NADH reducing cytochrome *c* through an alternate NADH oxidase pathway which has been reported previously in some cell types (Kerscher, 2000; Elguindy & Nakamaru-Ogiso,

2015). These results demonstrate that RIFS can be successfully used in frozen clinical blood samples.

## Discussion

Mitochondrial respiration plays a key role in metabolic, cardiovascular, and aging-related diseases. However, analyses measuring mitochondrial oxidative phosphorylation within physiological rates previously required immediate processing of viable tissue samples given the damage of the inner membrane to freeze–thaw. This requirement made respirometry analysis challenging for stored samples, biobanks, and long-term studies for age-related diseases. We have designed a specific combination of substrates and inhibitors that preserve maximal mitochondrial electron transport activity

in frozen biological specimens. Direct comparison confirmed that our frozen mitochondrial respirometry assay accurately detects differences in ETC activity and respirometry observed in fresh samples. Our novel formulation allows for simultaneous measurements of respiration driven by electron entry through Complex I and II, while determining Complex IV individual activity. The simplified sample preparation and the 96-well format of the Seahorse XF96 allow using significantly less biological material and do not require preparation of mitochondria or the use of detergents or permeabilizing agents. Furthermore, we demonstrated that differences in maximal mitochondrial electron transport capacity observed in fresh clinical samples from patients with pheochromocytoma (Vergnes et al, 2016) were maintained after freezing and multiple freeze–thaw cycles over a period of several years. This is particularly important in age-related studies since it enables a longitudinal design in which

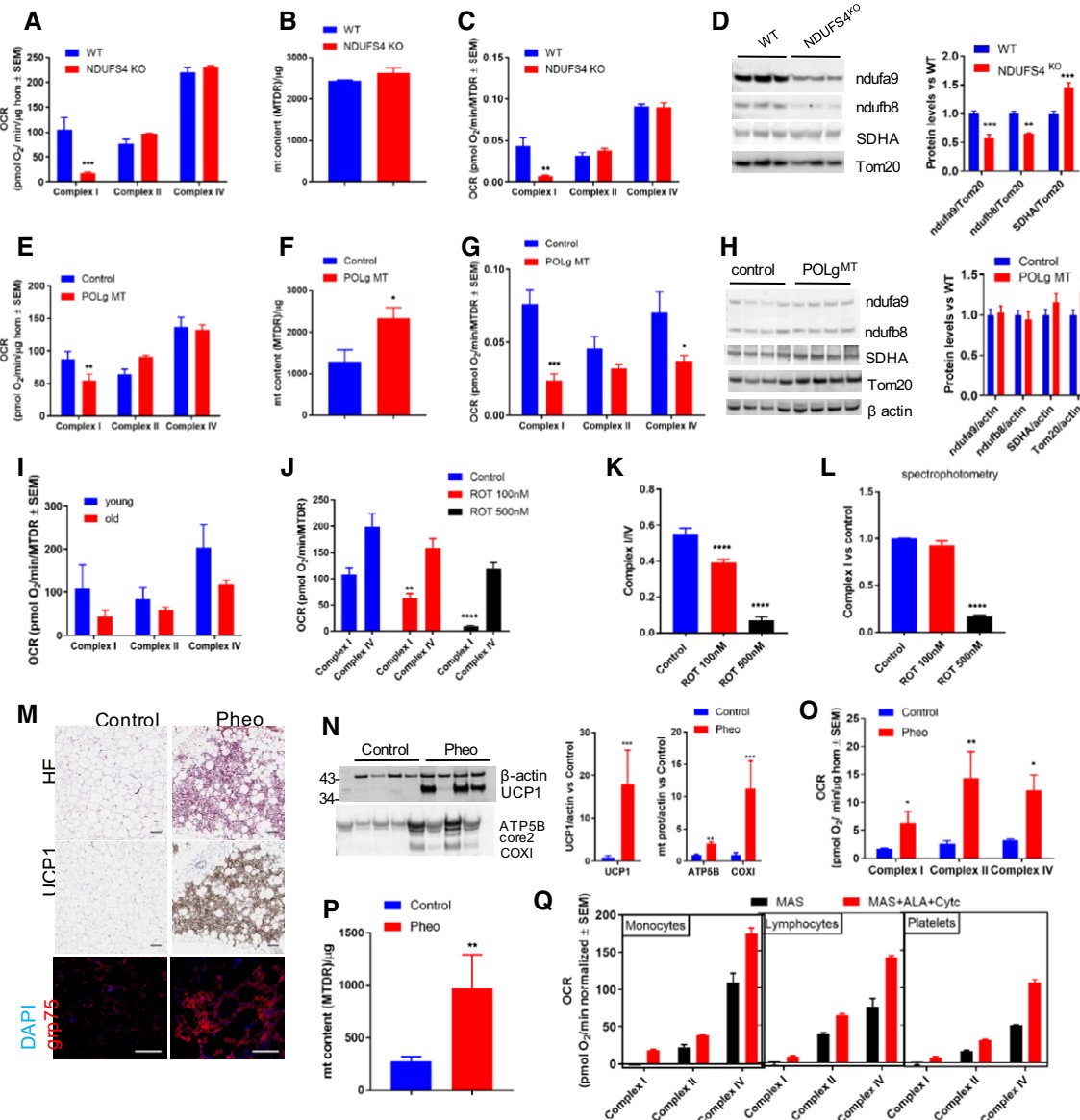

**Figure 7.**

**Figure 7. RIFS validation in pathophysiological relevant mouse and human disease models.**

A–D Complex I-, II-, and IV-dependent respiration by protein (A), MTDR mitochondrial content (B), Complex I-, II-, and IV-dependent respiration corrected by mitochondrial mass (C), and Western blot followed by quantification (D) in heart homogenates from WT and *Ndufs4* KO mice.

E Complex I-, II-, and IV-dependent respiration by protein in *Polg* mutant (*Polg* MT) and control heart homogenates.

F MTDR mitochondrial content in *Polg* MT and control heart homogenates.

G Complex I-, II-, and IV-dependent respiration corrected by mitochondrial mass.

H Western blot followed by quantification in heart homogenates from WT (control) and *Polg* MT mice.

I Quantification of Complex I-, II-, and IV-dependent respiration in zebrafish muscle homogenate from young and old fish.

J Quantification of Complex I- and IV-dependent respiration using RIFS in zebrafish deyolked embryo homogenate acutely treated with rotenone before freezing.

K Complex I to Complex IV ratio in zebrafish deyolked embryo homogenate in the indicated conditions.

L Complex I activity measure spectrophotometrically in zebrafish deyolked embryo homogenate in the indicated conditions.

M Representative H&E sections from human donors show the presence of multilocular adipose tissue in the pheochromocytoma (Pheo) sample. UCP1-DAB staining (central panels) and Grp75/mortalin fluorescence staining (bottom panels) from control and Pheo patients.

N Western blot followed by quantification in homogenates from control and Pheo patients.

O Complex I-, II-, and IV-dependent respiration by protein in periadrenal adipose tissue homogenates from control and Pheo human patients.

P Mitochondrial content by MTDR quantification in homogenates from control and Pheo human patients.

Q Quantification of complex activities (I, II, and IV) in MAS versus MAS supplemented with Cyt *c* and ALA buffer in isolated and cryopreserved human monocytes, lymphocytes, and platelets.

Data information: Biological replicates: (A–I), $n = 3$; (J–L), $n = 3$ (each $n$ represents a pool of 300 embryos); (N–P), $n = 8$; and (Q), $n = 3$–6 per group. Every biological replicate represents the average of at least four technical replicates. Data are the mean ± SEM. Western blots illustrate a representative blot of three independent experiments. The stainings are representative of eight control and four Pheo cases. Three representative fields were imaged and assessed for every biological sample. Contrast and brightness settings are the same across the samples. Scale bars represent 100 μm. For calculating statistics, the number of replicates used is the biological replicates indicated above. We used the following tests: two-way ANOVA with Sidak's multiple comparison test in panels (A–E, G, and O); one-way ANOVA Tukey's multiple comparison test in panels (J–L); unpaired *t*-test in panels (F, H, N, and P). *$P < 0.05$; **$P < 0.01$; ***$P < 0.001$; ****$P < 0.0001$.

Source data are available online for this figure.

bioenergetic health can be monitored over the several years it frequently takes to develop neurodegenerative diseases such as Alzheimer's disease.

We also reproduced data from heart homogenates of mitochondrial Complex I-deficient mice (Karamanlidis *et al*, 2013). Respiration in previously frozen tissue demonstrated the expected phenotype of impaired Complex I-dependent respiration in *Ndufs4* knockout mice. Our technology can also detect acute changes in mitochondrial respiratory function in response to compound treatment in previously frozen samples. Treatment with metformin and phenformin in previously frozen mitochondrial preparation demonstrated the expected dose-dependent decrease in Complex I-dependent respiration dependent through Complex I (Bridges *et al*, 2014). This is significant because it indicates that compounds can be screened for mitochondrial toxicity in a standard frozen mitochondria preparation that will increase the throughput for screening assays. Similarly, mitochondria have been proposed as a key target of environmental toxins. Screening of compounds for mitochondrial toxicity is limited by the requirement for freshly isolated mitochondria or intact cells that are grown in conditions that are not representative of physiological metabolic conditions. Our approach would allow for an efficient method to screen for environmental toxins in tissue samples collected in the field and thereby monitor the mitochondrial functional impact of environmental exposures within a large population (Meyer *et al*, 2013; Zolkipli-Cunningham & Falk, 2017).

Importantly, our simplified sample preparation requires less starting material and can be applied to cells in culture and those isolated from human blood, which opens the door to clinical-based research in bioenergetic health monitoring in large populations.

In summary, the RIFS method will therefore dramatically increase access to mitochondrial functional testing and catalyze new discoveries in science and medicine with great potential for retrospective research studies, clinical trials, and population exposure monitoring.

# Materials and Methods

### Mice

All animal procedures were performed in accordance with the Guide for Care and Use of Laboratory Animals of the NIH and were approved by the Animal Subjects Committee of the University of California, Los Angeles. Tissues were collected from 12-week-old male C57BL6/J mice (Jackson lab, Bar Harbor, ME). Animals were fed standard chow (mouse diet 9F, PMI Nutrition International, Brentwood, MO) and maintained under controlled conditions (19–22°C and a 14:10 h light–dark cycle) until euthanasia.

### Zebrafish husbandry and strains

Zebrafish (*Danio rerio,* Oregon AB) were housed at the Zebrafish facility of the School of Biology and Medicine and maintained at 28.5°C and on a 14:10 h light–dark cycle. Embryos were staged by hours or days post-fertilization according to previous publication (Kimmel *et al*, 1995). Young male zebrafish (1 year old) or old male zebrafish (2 year old) were used. Zebrafish husbandry and welfare was approved by the *Service de la consommation et des affaires vétérinaires* (SCAV) of the Canton of Vaud.

### Human subjects

All human study procedures were approved by the University of California, Los Angeles Institutional Review Board (IRB#13-001332-AM-00005) or the University of Alabama at Birmingham Institutional Review Board (Protocol #X110718014) for the isolation of cells from human blood from healthy subjects. Tumor adjacent retroperitoneal fat biopsies were obtained from pheochromocytoma patients or control patients with neoplasms as described previously (Vergnes *et al*, 2016).

## Mitochondrial and homogenate sample preparation for fresh versus frozen comparison

All procedures were performed using pre-chilled equipment and solutions. One C57BL6 mouse was used per isolation (biological replicate). The mouse was euthanized with isoflurane, and the peritoneal wall was opened with a U-shaped incision in the lower abdominal area.

### Mouse liver isolation

The liver was removed, washed with PBS (Sigma), and minced with scissors. To perform a comparison between fresh and frozen samples, half of the liver was snap-frozen in liquid nitrogen for 10 min before homogenization. Samples were placed in a Potter-Elvehjem (Teflon-glass) homogenizer (6–9 strokes) with 10 ml of ice-cold isolation buffer (250 mM mannitol, 75 mM sucrose, 100 μM K-EDTA, 10 mM KHEPES, pH 7.4) supplemented with 500 μM K-EGTA (pH 7.4). Homogenates were centrifuged at 1,000 $g$ for 10 min at 4°C; then, the supernatant was removed and re-centrifuged at 1,000 $g$ for 10 min at 4°C. We collected 1 ml of the resulting supernatant as the homogenate sample, and the rest was centrifuged at 10,000 $g$ for 10 min at 4°C. The mitochondrial pellets were washed twice in wash buffer (isolation buffer supplemented with 0.5% fatty acid-free BSA, MilliporeSigma). The final mitochondrial pellet was re-suspended in ice-cold isolation buffer with no BSA. BSA was omitted from the final isolation buffer to prevent interference with the protein assay kit (BCA, Thermo Fisher).

### Mouse white adipose tissue isolation

Gonadal WAT was harvested and rinsed in PBS. Tissue was minced and suspended in 2 ml (~ 1 ml/500 mg tissue) of sucrose–HEPES–EGTA buffer supplemented with BSA (SHE + BSA; 250 mM sucrose, 5 mM HEPES, 2 mM EGTA, 2% fatty acid-free BSA, pH 7.2). The preparation was then mechanically homogenized with nine strokes in a glass–glass Dounce homogenizer. Homogenates were centrifuged at 1,000 $g$ for 10 min at 4°C; then, the supernatant was removed and re-centrifuged at 1,000 $g$ for 10 min at 4°C. We collected 1 ml of the resulting supernatant as the homogenate sample, and the rest was centrifuged at 10,000 $g$ for 10 min at 4°C. Mitochondrial pellet was re-suspended in SHE + BSA and centrifuged with the same settings once more. The final mitochondrial pellet was then re-suspended in SHE without BSA. BSA was omitted from the final isolation buffer to prevent interference with the BCA assay.

## Tissue homogenization from frozen samples

Frozen tissues were thawed in ice-cold PBS, minced, and homogenized in MAS buffer (70 mM sucrose, 220 mM mannitol, 5 mM KH$_2$PO$_4$, 5 mM MgCl$_2$, 1 mM EGTA, 2 mM HEPES pH 7.4). Heart and WAT were mechanically homogenized with 20 strokes in glass–glass Dounce homogenizer. Liver, BAT, brain, kidney, lung, and muscle were mechanically homogenized with 10–20 strokes (depending on the tissue: lung and muscle require larger number of strokes) in Teflon-glass homogenizer. All homogenates were centrifuged at 1,000 $g$ for 10 min at 4°C; then, the supernatant was collected. Protein concentration was determined by BCA (Thermo Fisher).

For muscle homogenization, after mincing the tissue, trypsin (0.25 mg/ml final concentration, Sigma) or collagenase Type II (0.25 mg/ml final concentration, Worthington) was added to MAS buffer and samples were incubated at 37°C for 10 or 30 min prior to centrifugation. Based on protein supercomplex integrity analysis by BNGE, collagenase treatment for 30 min at 37°C was the optimal treatment to provide higher respiration rates while keeping the inner mitochondrial membrane intact.

Frozen zebrafish muscle tissues were thawed in ice-cold PBS, minced, and homogenized in MAS (70 mM sucrose, 220 mM mannitol, 5 mM KH$_2$PO$_4$, 5 mM MgCl$_2$, 1 mM EGTA, 2 mM HEPES pH 7.4). After mincing the muscle tissue, collagenase Type IV (1 mg/ml final concentration) was added to MAS buffer and samples were incubated at 28°C for 30 min prior to centrifugation. Muscle tissue was mechanically homogenized with 10 strokes in glass–glass Dounce homogenizer. Homogenates were centrifuged at 1,000 $g$ for 10 min at 4°C; then, the supernatant was collected to assess protein concentration.

Dechorionated embryos at 48 h post-fertilization were treated with 100 nM or 500 nM of rotenone directly delivered in eggs water for 2 h 45 min. Deyolked embryos were snap-frozen in liquid nitrogen. Frozen embryos were thawed in MAS and were mechanically homogenized with 10 strokes in glass Dounce homogenizer. Homogenates were centrifuged at 1,000 $g$ for 10 min at 4°C; then, the supernatant was collected.

## Human monocyte cell line and human derived blood cells

In order to assess RIFS protocol in blood cells (monocytes, lymphocytes, and platelets), first the conditions were optimized with the monocytic cell line THP-1. The THP-1, leukemic monocyte cells (Invitrogen, Cat# A10491-01) were grown in RPMI-1640 1X (Invitrogen-ATCCC modification) supplemented with fetal bovine serum (10%), 0.05 mM β-mercaptoethanol, and penicillin/streptomycin. The cells were counted and divided into two aliquots to achieve 80,000 cells/30 μl per well, one of the aliquots was dry pelleted and frozen in liquid nitrogen, and after 30-min cryopreservation the vial was thawed and suspended in MAS buffer. The isolation of monocytes, lymphocytes, and platelets was performed by established methods in the Darley-Usmar lab (Chacko *et al*, 2013; Kramer *et al*, 2014). After the isolation, the cells were centrifuged at 700 $g$ for 5 min at room temperature, and the pellet was frozen in liquid nitrogen and cryopreserved.

## Respirometry assay using conventional Seahorse protocol

Mitochondria and homogenates were loaded into Seahorse XF96 microplate in 20 μl of MAS containing substrates. The loaded plate was centrifuged at 2,000 $g$ for 5 min at 4°C (no brake), and an additional 130 μl of MAS + substrate was added to each well. To avoid disrupting mitochondrial adherence to the bottom of the plate, we added MAS using multichannel pipette pointed at a 45° angle to the top of well chamber, as instructed by the manufacturer. Substrate concentrations were as follows: 5 mM pyruvate + 5 mM malate, 1 mM NADH, or 5 mM succinate + 2 μM rotenone. Substrates plus ADP were injected at port A (3.5 mM final concentration), oligomycin at port B (3.5 μM), FCCP at port C (4 μM), and antimycin A at port D (4 μM). Mix and measure times were 0.5 and 4 min,

respectively. A 2-min wait time was included for oligomycin-resistant respiration measurements.

Alternatively, when mitochondria started in state 1 we injected: (i) 5 mM pyruvate + 5 mM malate or 5 mM succinate + 2 μM rotenone at port A; substrates plus ADP at port B (3.5 mM final concentration); CAT at port C (3.5 μM); and azide at port D (50 mM) or (ii) substrates plus ADP at port A (5 mM pyruvate + 5 mM malate or 5 mM succinate + 2 μM rotenone plus ADP 3.5 mM); CAT at port B (3.5 μM); $N,N,N',N'$-tetramethyl-$p$-phenylenediamine (TMPD) + ascorbic acid (0.5 mM + 1 mM) at port C; and azide at port D (50 mM).

For liver mitochondria, we loaded 6 μg for pyruvate + malate and NADH and 3 μg for succinate + rotenone-dependent respiration. For liver homogenate, we loaded 15 μg for pyruvate + malate and NADH and 8 μg for succinate + rotenone-dependent respiration.

### Respirometry assay using RIFS

Mitochondria and homogenates were loaded into Seahorse XF96 microplate in 20 μl of MAS. The loaded plate was centrifuged at 2,000 $g$ for 5 min at 4°C (no brake) and an additional 130 μl of MAS for mitochondria or MAS containing cytochrome $c$ (10 μg/ml, final concentration), in the case of homogenates, was added to each well with the same consideration as in the fresh protocol. In the case of brain and lung homogenates, alamethicin (10 μg/ml, final) was also added to the MAS containing cytochrome $c$ solution to allow complete membrane permeabilization to substrates. Substrate injection was as follows: pyruvate + malate (5 mM each), NADH (1 mM), or 5 mM succinate + rotenone (5 mM + 2 μM) were injected at port A; rotenone + antimycin A (2 μM + 4 μM) at port B; TMPD + ascorbic acid (0.5 mM + 1 mM) at port C; and azide (50 mM) at port D. These conditions allow for the determination of the respiratory capacity of mitochondria through Complex I, Complex II, and Complex IV.

When using RIFS in tissue lysates, we loaded the same protein amount independently of the substrate. For liver, we loaded 4 and 8 μg for mitochondrial and homogenate samples, respectively. For WAT, we loaded 6 and 15 μg for mitochondrial and homogenate samples, respectively. We loaded the following protein homogenates for BAT (3 μg), heart (2 μg,), kidney (4 μg), brain (4 μg), skeletal muscle (6 μg), soleus (6 μg), and lung (15 μg).

We used frozen liver mitochondria to test OXPHOS inhibitors specificity and ATP allosteric inhibition of COX. Inhibitors (metformin, phenformin, 3-nitropropionic acid, and potassium cyanide) or ATP at the indicated concentration were added to MAS after centrifugation of the plate containing the mitochondria.

When using RIFS in zebra fish muscle homogenates, 30 μg of homogenates was loaded into Seahorse XFe24 microplate in 50 μl of MAS. When using RIFS in deyolked zebra fish embryos (pool of 300 embryos per condition, 30 μg of homogenate loaded per well). The loaded plate was centrifuged at 2,000 $g$ for 5 min at 4°C (no brake), and an additional 475 μl of MAS containing cytochrome $c$ (10 μg/ml, final concentration) was added to each well. Substrate injection was as follows: NADH (1 mM) or 5 mM succinate + rotenone (5 mM + 2 μM) were injected at port A; rotenone + antimycin A (2 μM + 4 μM) at port B; TMPD + ascorbic acid (0.5 mM + 1 mM) at port C; and azide (50 mM) at port D. These conditions allow for the determination of the respiratory capacity of mitochondria

through Complex I, Complex II, and Complex IV. The experiment was performed at 28°C.

When using RIFS in human cell lines, fresh and frozen THP-1 cells were seeded into a Seahorse XF96 plate at 80,000 cells/30 μl/well. The plate was centrifuged at 2,300 $g$ for 5 min with no brake. After centrifugation, 150 μl of MAS (fresh cells) or MAS supplemented with cytochrome $c$ (10 μM) and alamethicin 2.5 μg/ml (frozen) were added to each well. The cartridge was loaded with desire substrates, for Complex I determination a mix of PMP + pyrmal + FCCP (fresh) or NADH (frozen). Complex II substrates were PMP + succinate + rotenone + FCCP for both fresh and frozen cells. Then, rotenone (2 μM) or antimycin A (10 μM) to inhibit Complex I or Complex II, followed by ascorbate–TMPD for Complex IV activity and finally azide 20 mM was injected.

The isolated blood cells were thawed and suspended in XF-DMEM media to seed 150,000 monocytes/well, 300,000 lymphocytes/well, and $10 \times 10^6$ platelets/well in 30 μl media. The plate was centrifuged at 1,300 $g$ for 1 min with no brake, then rotated by 180 degrees and centrifuged again. After centrifugation, 150 μl of MAS or MAS supplemented with cytochrome $c$ (10 μM) and ALA 2.5 μg/ml (frozen) were added to each well. The cartridge was loaded with same concentrations of substrates and inhibitors used for THP-1 cells.

### Seahorse data analysis

Wave software (Agilent) was used to export OCR rates normalized by protein to GraphPad Prism v7.02. Complex I-, II-, and IV-dependent respiration was calculated by subtracting OCR values from the substrates subtracted from the ones from the inhibitors.

### Spectrophotometrical enzymatic activities

Complex I activity was measured in isolated mitochondria and in tissue lysates following NADH oxidation at 340 nm as described previously (Garaude *et al*, 2016). CS activity was performed in tissue lysates using 5,5′-dithiobis(2-nitrobenzoic acid) (DTNB) as described (Benador *et al*, 2018).

### Solution preparation

All solutions are prepared with the highest purity deionized water or ultrapure dimethyl sulfoxide (DMSO, VWR). pH in all aqueous solutions was adjusted to 7.2 using potassium hydroxide (KOH) and hydrochloric acid (HCl). These solutions can be prepared and stored at −20°C: cytochrome $c$: 100 μg/ml MAS; alamethicin: 20 mg/ml in ethanol; pyruvate: 0.5 M in MAS; malate: 0.5 M in MAS; succinate: 0.5 M in MAS; rotenone: 40 mM in DMSO; antimycin A: 20 mM in DMSO; and azide: 0.5 M in MAS. These solutions need to be prepared fresh the day of the assay: 10 mM NADH in MAS, TMPD + ascorbate: 10 mM ascorbate in MAS (adjusted to pH 7.4), 5 mM TMPD in 10 mM ascorbate.

### Protein gel electrophoresis and immunoblotting

#### SDS–PAGE

2.5–10 μg of mitochondria or 5–20 μg of total lysate was loaded into 4–12% Bis-Tris precast gels (Thermo Fisher Sci. NP0321), and gel

electrophoresis was performed in xCell SureLock (Novex) under constant voltage at 80 V for 15 min (to clear stacking) and 120 V for 60 min.

### Blue native gel electrophoresis

Twenty-five microgram of muscle homogenates protein was re-suspended in 20 μl solubilization buffer (50 mM imidazole, 500 mM 6-aminohexanoic acid, EDTA 1 mM pH 7.0) (Wittig *et al*, 2006). 1 mg digitonin/mg of mitochondrial protein was added, and samples were incubated on ice for 5 min. One percent digitonin (Sigma) was dissolved in PBS by boiling and stored at 4°C until use. Solubilized samples were centrifuged at maximal speed in a microcentrifuge (Thermo Fisher) for 30 min at 4°C. Pellet was discarded, and supernatant was combined with 1 μl of 2.5% Coomassie G-250. Samples were loaded into NativePAGE 3–12% Bis-Tris gel and electrophoresed at 4°C in xCell SureLock (Novex) in constant voltage at 20 V for 60 min and 200 V for 120 min or until dye front exited the gel.

### Immunoblotting

Proteins were transferred to methanol-activated PVDF membrane in xCell SureLock under 30 V constant voltage for 1 h at 4°C. Coomassie was completely washed off blue native blots using 100% methanol. Blots were blocked with 3% BSA in PBST (1 ml/l Tween-20/PBS) and incubated with primary antibody diluted in 1% BSA/PBST overnight at 4°C. The next day, blots were washed 3 × 10 min in PBST, probed with IRDye 800CW and IRDye 680RD secondary antibodies diluted in blocking solution for 1 h at room temperature, and rinsed again 3 × 10 min in PBST. Detection was achieved using ChemiDoc Molecular Imager (Bio-Rad). Band densitometry was quantified using ImageJ Gel Plugin (NIH). We used the following antibodies: mtOXPHOS cocktail (for combined ATP5B, core2, COXI, and SDHB detection-ab110413), ndufa9 (ab188373), UCP1 (ab10983), and β-actin (ab8227) from Abcam; ndufb8 (459210) and SDHA (459200) from Thermo Fisher Biotechnology; core2 (14742-1-AP) from Proteintech; TOM20 (Sc-11415) from Santa Cruz Biotechnology; and vinculin (V9131) from Sigma-Aldrich.

### Mitochondrial quantification using MitoTracker Deep Red

Fresh mitochondria (5 μg, per condition) were incubated in MAS plus substrates (pyruvate + malate, 5 mM each) in the presence of FCCP (1 μM, final) or calcium chloride (0.01–2 mM, final) for 10 min at 37°C. Mitochondria were centrifuged at 1,000 $g$ for 10 min and re-suspended in MAS plus substrates containing either MTDR (1 μM, final) or MTR (1 μM, final) for 10 min at 37°C. Dye was removed by centrifuging the mitochondria at 1,000 $g$ for 10 min. The stained mitochondrial pellet was re-suspended in 100 μl of MAS and fluorescence measured in clear-bottom black 96-well plate (Corning). MTDR was excited at 625 nm and its emission recorded at 670 nm. MTR was excited at 581 nm and its emission recorded at 644 nm. Fluorescence was plotted as percentage of the control, untreated mitochondria.

For MTDR mitochondrial content quantification, 2.5–80 μg of mitochondrial or lysate samples was seeded per well on a clear-bottom black 96-well plate in 100 μl of MAS containing MTDR (1 μM, final) and incubated at 37°C for 10 min. Plates were centrifuged at 2,000 $g$ for 5 min at 4°C (no brake), and supernatant was carefully removed. Finally, 100 μl of MAS was added per well and MTDR

fluorescence measured as indicated before. Mitochondrial content was calculated at MTDR signal (minus blank) per milligram of protein. Alternatively, plates were imaged using the Operetta system.

### Cell culture and imaging

INS-1 832/13 (INS1) were cultured in RPMI 1640 medium supplemented with 10% fetal calf serum (FBS), 10 mM HEPES buffer, 1 mM pyruvate, 50 mM 2-β-mercaptoethanol, 50 U/ml penicillin, and 50 mg/ml streptomycin in 37°C, 5% $CO_2$. HepG2 cells were cultured in DMEM (25 mM glucose) supplemented with 10% FBS, 50 U/ml penicillin and 50 mg/ml streptomycin, and 10 mM HEPES buffer in 37°C, 5% $CO_2$. For the determination of the retention of MTR CMXRos or Deep Red FM after fixation, and capacity of the dyes to stain fixed cells, cells were grown on Greiner CELLview™ 4-compartment dishes with glass-bottom. INS1 cells were incubated in 100 nM of dye in complete media for 30 min and subsequently washed with complete media once before imaging on a Zeiss LSM880 confocal laser scanning microscope with a 63× Apochromat oil-immersion objective ("before fixation") in Airyscan mode. After the first round of imaging, cells were fixed with 4% paraformaldehyde for 30 min on stage, washed three times with 1× PBS, pH 7.4, and imaged again ("4% PFA"). Subsequently, cells were incubated in PBS with 0.1% Triton X-100 for 5 min, washed with PBS three times, and imaged again ("0.1% Triton X-100 wash"). HepG2 cells were incubated in 200 nM of dye in complete media for 40 min and washed with complete media before imaging ("before fixation"). A separate dish with HepG2 cells was fixed with 4% PFA for 30 min before staining with 200 nM of dye. Cells were then washed with 1× PBS, pH 7.4 three times and imaged ("after fixation").

### Histology

Human fat biopsies were fixed overnight in 10% neutral buffered formalin solution before being dehydrated and embedded in paraffin. Deparaffinized 4-μm sections were stained with hematoxylin and eosin. Slides were scanned using an Aperio AT high-throughput scanning system.

### UCP1 immunohistochemistry

UCP1 levels in 4 μm human fat sections were assessed as previously described (Vergnes *et al*, 2016) using a UCP1 primary antibody (Abcam, ab10983) at 1:500 dilution. Sections were counterstained with hematoxylin. Slides were scanned using an Aperio AT high-throughput scanning system.

### Mitochondrial immunofluorescence

Heat-induced antigen retrieval (HIER) was carried out for all sections in 1 mM EDTA buffer, pH 8.00 using a Biocare Decloaker at 95°C for 25 min. Autofluorescence was blocked by incubation with 0.1% Sudan Black B in methanol followed by two washes with 70% ethanol and PBS. Slides were quenched in 50 mM glycine–PBS for 10 min and subsequently blocked with 2% BSA-PBS for 30 min. Mitochondrial staining was performed using a grp75/mortalin antibody (Antibodies Inc, 75–127) at 1:1,000 dilution overnight at 4°C. The secondary antibody was Alexa Fluor 647 donkey anti-rabbit, and nuclei were stained with DAPI. Sections were imaged with a Zeiss LSM 880 confocal microscope using 405 and 633 nm laser excitation.

## Statistical analysis

Comparisons between groups were made by one-way or two-way analysis of variance (ANOVA). Correction for multiple comparisons was made by Tukey or Sidak's tests when appropriate. Pairwise comparisons were made by two-sided Student's *t*-test. Differences were considered statistically significant at $P < 0.05$. In the figures, asterisks denote statistical significance (*$P < 0.05$; **$P < 0.01$; ***$P < 0.001$; ****$P < 0.0001$). Data were analyzed with GraphPad Prism v7.02. In the figures, each point represents a biological replicate.

# Data availability

There are no primary datasets produced in this study.

Expanded View for this article is available online.

## Acknowledgements

The authors would like to acknowledge Drs Masha Livhits and Michael Yeh for sharing human sample material. The authors would like to thank Jennifer Ngo for help with manuscript editing, writing, and drawings. The authors would like to thank Alexandra Brownstein and Corey Osto as well as Drs. Michael Shum, Daniel Dagan, Evan Taddeo, Alex Van der Bliek, Jose Antonio Enriquez, and Amy Wang for scientific discussions and valuable advice. This work was supported by UAB-UCSD O'Brien Center P30 DK079337 (VMDU), UAB Nathan Shock Center P30 G050886 (VMDU), HL110349 (RT), U54 DK120342 (KR and LV), R56AG060880 (JW), R01AG055518 (JW), K02AG059847 (JW), R21AR072950 (JW), UCLA Older Americans Independence Center P30AG028748 (JW), R01 DK099618-05 (OSS), R01 CA232056-01 (OSS), R21AG060456-01 (OSS), R21 AG063373-01 (OSS), ADA Grant No. 1-19-IBS-049 (OSS), R01AA026914-01A1 (ML), and Swiss National Science Foundation Grant No. 320030_170062 (FA).

## Author contributions

Conceptualization: RA-P, IYB, LS, OSS; Methodology: RA-P, IYB, LS, GAB, FA; Validation: RA-P, SL, AC, LV, FA, GK, RT, KR, JW, HS; Investigation: RA-P, IYB, AP, MV, GAB, SL, FA, AC, GK, VMD-U; Human sample acquisition: LV, KR, HS; Zebrafish studies: SL, FA; Writing—original draft: RA-P, IYB; Writing—review & editing: RA-P, IYB, AP, GAB, MV, AC, LV, AM, RT, KR, HS, JW, FA, VMD-U, ML, AD, LS, OSS; Funding acquisition: FA, VMD-U, ML, LS, OSS; Resources: FA, VMD-U, ML, LS, OSS; Supervision: RA-P, FA, VMD-U, LS, OSS.

## Conflict of interests

The authors declare that they have no conflict of interest.

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
