## [Review Process File · The EMBO Journal]

A NOVEL APPROACH TO MEASURE MITOCHONDRIAL RESPIRATION IN PREVIOUSLY FROZEN BIOLOGICAL SAMPLES

Rebeca Acin-Perez, Ilan Y. Benador, Anton Petcherski, Michaela Veliova, Gloria A. Benavides, Sylviane Lagarrigue, Arianne Caudal, Laurent Vergnes, Anne Murphy, Georgios Karamanlidis, Rong Tian, Karen Reue, Jonathan Wanagat, Harold Sacks, Francesca Amati, Victor M. Darley-Usmar, Marc Liesa, Ajit Divakaruni, Linsey Stiles and Orian S. Shirihai

Review timeline:

Submission date:	25 th November 2019
Editorial Decision:	20 th December 2019
Revision received:	30 th January 2020
Editorial Decision:	1 st March 2020
Revision received:	11 th March 2020
Accepted:	19 March 2020

Editor: Elisabetta Argenzio

Transaction Report:

1st Editorial Decision

20th December 2019

Thank you for submitting your resource article entitled "A novel approach to measure mitochondrial respiration in previously frozen biological samples" [EMBOJ-2019-103889] to The EMBO Journal. Your study has been sent to three referees for evaluation, whose reviews are enclosed below.

As you can see, while referee #1 and #2 find the presented method potentially interesting, they raise several points that have to be addressed before they can support the publication of your work in The EMBO Journal. In light of referee #3's comments, I informed him/her that the journal publishes novel methods that clearly advance the field and asked to reweigh the manuscript. While this referee recognizes the potential value of the manuscript, s/he requests you to modify the title and to address the concerns related to the measurement of oxygen consumption listed in his/her report below.

Given the overall interest of your study, I would like to invite you to revise the manuscript in response to the referee reports. I should note that conclusively addressing these and all the other referees' points is essential for publication in The EMBO Journal.

REFeree REPORTS

Referee #1:

This manuscript, authored by renowned researchers in the field of mitochondrial biology, examines

a proposes a novel approach to measure mitochondrial respiratory chain function by respirometry in frozen tissue homogenates. Hereto forth oxygen consumption was thought to be reliable only in mitochondria isolated from fresh samples. By manipulating substrates, additives and modes of permeabilization the authors showed that informative data can also be obtained from homogenates obtained from a small amount of various frozen tissues. These findings could be of practical relevance in clinical settings allowing storage and transport of frozen tissue rather than, the often problematic immediate workup of fresh tissue.

My main concerns are twofold ;

1-The freezing procedure of tissues was done in an optimized laboratory setting and not in a possibly problematic, clinical environment.
For example ; tissues obtained in the OR/clinic are often transported at room temp to a freezer -20C? -80C? for storage. Not every hospital has access to bedside liquid nitrogen.

It would be interesting to know if/how much respiratory capacity would be preserved if the sample would remain 30min in room temp, than stored in the freezer prior to shipping (dry ice? Ice blocks?). This could also depend on the tissue, liver might be more vulnerable than muscle.

2-Possible pitfalls and drawbacks are sometimes not adequately addressed.
examples

- a- How could a defect in PDHc/Krebs cycle defect behave in this setting?
- b- How would you identify an isolated CIII defect?
- c- Would you deal with a CoQ defect?
- d- Could haemoglobin (i.e. RBC contamination in tissue homogenate interfere? for examples with MTDR measuring mitochondrial content?
- e- Are there other limitations that should be mentioned?

Minor comments;

Conventional methods are sometimes not accurately related to; for example mitochondrial isolation can easily be done in ~0.2g muscle -fresh or frozen- and without trypsin (the manuscript states 0.5-2g with trypsin).

When/if/how would, conventional methods be needed to verify oxygen consumption results (see former comments) ?

Referee #2:

The manuscript submitted by Acin-Perez and colleagues describes a method to measure oxygen consumption in frozen biological samples. According to the authors, although the integrity of the inner mitochondrial membrane is lost during freezing as we already know, the complexes of the electron transport chain are still intact and functional. Therefore, providing substrates that directly feed the ETC complexes should be enough to elicit electron flux and consume oxygen. Importantly, this approach enables the analysis of maximal uncoupled mitochondrial respiration in previously frozen samples such as patient-derived tissue from biobanks. The underlying idea is logic and simple. Eventually, analysis of (functional) supercomplexes can be carried out both in fresh and frozen samples with minimal problems, and the same should apply to oxygen consumption. Accordingly, while reading this manuscript I was asking why I didn't think of this idea myself. Overall, data are well performed and discussed, and most of the necessary controls are already in place. While data on OCR are strong, the section on mitochondrial quantification using a fluorescent indicator is rather weak and does not add so much to the overall story. Some points need more attention:

- I do not understand the difference in basal OCR in some panels, including 1D, 1G, 3C and 3E. Basal respiration should correspond to state 1 (no substrates, no ADP) and no difference between fresh and frozen samples should be present. Indeed, according to the figures and legends, both substrates and ADP are injected using the first port of seahorse reader. Therefore, traces should begin (i.e. first time point) with comparable OCR. I think this is simply due to erroneous labeling of

the figures. Most likely, substrates (pyr/mal in panel 1A, succ/rot in panel 1D and NADH in panel 1G) are already present at the beginning of the experiment (state 4 is thus measured here), and only ADP is added with the first injection. Please fix this problem and provide control experiments showing state 1 respiration.

- As far as I know, Chance and Williams define state 2 respiration as a condition with high ADP and low substrates. Conversely, figure legend 1 reports "state 2 (substrate without ADP)", that should be instead considered as state 4. Accordingly, respiratory control ratios (RCRs) are defined as $\text{state3}/\text{state4}$ (not $\text{state3}/\text{state2}$ as shown).
- As an additional control, the authors should test the effect of atractyloside. It should inhibit state 3 respiration in fresh but not in frozen samples.
- The least convincing part of this work is the use of Mitotracker DeepRed to normalize mitochondrial content. I see several problems here. 1) Figure 6A demonstrates that some parameters can impact on MTDR fluorescence, since Ca^{2+} treatment causes an approximately 25% decrease of fluorescence with no apparent explanation. 2) Figure 6B clearly shows that the relationship between protein content and MTDR fluorescent is linear only over a narrow range (up to 10 μg for isolated mitochondria, without considering that the linearity in the case of total homogenate looks questionable). In a real-life situation, it would be difficult to know if MTDR signal is at saturation or not only looking at raw fluorescence. 3) When comparing figure 7F to 7H, MTDR vs protein show very different results. According to MTDR signal, mitochondrial content is twice higher in the POLg sample, i.e. an effect that should be well visible with other techniques. According to Western blot, the increase in POLg is very limited, thus questioning results obtained with MTDR. Overall, I think the main problem here is that we don't know exactly how MTDR works, and it is thus difficult to understand its strengths and limitation. I think this part can be omitted with no impact on the overall conclusions.

Referee #3:

This paper describes measurements of mitochondrial respiratory chain activity from various tissues and biological sources. The aim is purely methodological: to demonstrate that activity is maintained despite freezing and thawing the samples, which obviously saves a lot of material. My major concern is that there is no actual conceptual novelty in this work; its value may be considerable but it is purely methodological. This is more a matter of decision by the Journal's Editors. To my mind this work is not suitable for the EMBO Journal but should be published in a methodological forum.

I have somewhat minor concerns related to the method of measuring oxygen consumption. There is no information of how the actual numbers of O_2 consumption were obtained. What was the temperature (oxygen solubility is highly temperature dependent)? How was the oxygen concentration determined? How did the authors control for oxygen diffusion into the microplate wells? How was the zero oxygen point established?

Referee #1:

This manuscript, authored by renowned researchers in the field of mitochondrial biology, examines and proposes a novel approach to measure mitochondrial respiratory chain function by respirometry in frozen tissue homogenates. Heretofore oxygen consumption was thought to be reliable only in mitochondria isolated from fresh samples. By manipulating substrates, additives and modes of permeabilization the authors showed that informative data can also be obtained from homogenates obtained from a small amount of various frozen tissues. These findings could be of practical relevance in clinical settings allowing storage and transport of frozen tissue rather than, the often problematic immediate workup of fresh tissue.

My main concerns are twofold ;

1-The freezing procedure of tissues was done in an optimized laboratory setting and not in a possibly problematic, clinical environment.

For example ; tissues obtained in the OR/clinic are often transported at room temp to a freezer -20C? -80C? for storage. Not every hospital has access to bedside liquid nitrogen.

It would be interesting to know if/how much respiratory capacity would be preserved if the sample would remain 30min in room temp, than stored in the freezer prior to shipping (dry ice? Ice blocks?). This could also depend on the tissue, liver might be more vulnerable than muscle.

We agree with some of the reviewer concerns. In Figure 7 of the paper we have shown data from human samples (retroperitoneal fat from pheochromocytoma patients) obtained from the clinic and stored at -80 °C for 2 years. We are currently running an extended human study involving fat samples obtained from : i) periadrenal, ii) superclavicular, iii) para-/epicardial, and iv) subcutaneous depots. In this study, samples are collected in a clinical setting from the OR. Samples are collected from the patient and placed in ice cold PBS from 1-3 hours. Upon completion of the surgery, we pick up the samples and flash freeze them before storing them at -80 °C. We have successfully tested these samples for frozen respirometry. However, the reviewer is right in his/her concern about how much of the original activity is preserved, if the samples are kept in PBS at 4 °C or room temperature for an extended amount of time. To address that question, we have now run an experiment in mouse liver samples (3 independent samples), where we have collected samples from the same liver and processed them under different conditions:

- i) ice cold PBS for 30 min and freeze (4 °C 0.5h),
- ii) ice cold PBS for 3 hrs and freeze (4 °C 3h),
- iii) PBS at RT for 30 mins and freeze (RT 0.5h),
- iv) PBS at RT for 3 hrs and freeze (RT 3h) and
- v) freeze (LN)

The freezing step is done in liquid nitrogen for all the samples. Samples are then stored at $-80\text{ }^{\circ}\text{C}$ for at least 2 days before assessing how much of the respiratory capacity is preserved. The results below show that respiratory capacity is preserved when the samples are placed in ice cold PBS (the current procedure that we are following when collecting samples for the OR). When samples are stored at RT for extended amounts of time, Complex IV activity is fully preserved but Complex I activity is slightly decreased while Complex II is slightly increased, possible due to supercomplex rearrangements and/or respirasome instability. These results have been incorporated in Figure S6 (panels B and C) of the revised version of the manuscript.

2-Possible pitfalls and drawbacks are sometimes not adequately addressed. examples

a- How could a defect in PDHc/Krebs cycle defect behave in this setting?

We would not expect to observe defects of the TCA cycle, since we are adding exogenous NADH and succinate, both of which bypass TCA cycle and donate electrons directly to Complex I and Complex II, respectively. The limitation is related to the fact that we are measuring electron transport chain function capacity, rather than the integration of fuel oxidation and ETC activity, as previously discussed

b- How would you identify an isolated CIII defect?

We have now incorporated an additional assay to obtain an independent measure of complex III using duroquinol. Here, we are presenting the results of such measurements using heart homogenates from young and old mice.

c- Would you deal with a CoQ defect?

We have not tested CoQ deficient samples but we assumed that those defects will be picked up by our analysis because the RIFS assay, unlike the spectrophotometrical approaches, does not add any exogenous CoQ. Samples in RIFS are using their endogenous pool of CoQ that is mobilized when ETC is provided with either NADH or Succinate to Complex I and II, respectively.

d- Could haemoglobin (i.e. RBC contamination in tissue homogenate interfere? for examples with MTDR measuring mitochondrial content?

Once the tissue is collected from the subject, it is placed in PBS before freezing and most of the blood is washed out; therefore, we do not expect to have significant amounts of hemoglobin that would to interfere with MTDR mitochondrial content reading. Nevertheless, we have run an experiment where we include serial dilutions of MTDR, blood and blood plus MTDR. The results below show that RBC at levels that are found in tissues does not interfere with MTDR fluorescence. Hemoglobin levels in human blood is 12-17 g/dl. We added different volumes of human blood to a plate (0.01, 0.05, 0.1, 0.5, 1, 5, 10 and 20 μ l) and run the same protocol that we use to measure MTDR fluorescence. We observed that at higher concentrations, RBCs a limited amount of background (panels A and insert of panel B). However, when we prepare the lysates from tissues, we do not expect the contamination observed using 5-20 μ l of blood. We also measured fluorescence at increasing concentrations of MTDR and MTDR plus 1 μ l of blood and observed similar results (panel C). Note we have used the same Y scale axis in B and C for comparative purposes. These results are now part of the new Figure S5 (panels E and F).

e- Are there other limitations that should be mentioned?

We have tried RIFS in a variety of different tissues in mouse and human, cell lines and blood cells. The protocol is very straightforward in almost all cases with the exception of tissues that have a lot of fiber and collagen such as skeletal muscle or tissues that are very rich in membranes such as brain. In the latter cases, an optimization step was required that included either enzymatic digestion (skeletal muscle) or additional membrane permeabilization (brain) for the reagents and electron donors, particularly NADH, to be accessible to mitochondria.

Minor comments;

Conventional methods are sometimes not accurately related to; for example mitochondrial isolation can easily be done in ~0.2g muscle -fresh or frozen- and without trypsin (the manuscript states 0.5-2g with trypsin).

When/if/how would, conventional methods be needed to verify oxygen consumption results (see former comments)?

We now show lysates obtained from 20mg of tissue, an amount that would not be possible to isolate mitochondria. We agree that trypsin can be avoided to isolate mitochondria from muscle in traditional protocols, as stated by the reviewer. However, using collagenase Type II (0.25 mg/ml final concentration) in such a small amount of tissue to obtain lysates facilitates the access of NADH and the different fuels to mitochondria, as mentioned in the previous point.

Referee #2:

The manuscript submitted by Acin-Perez and colleagues describes a method to measure oxygen consumption in frozen biological samples. According to the authors, although the integrity of the inner mitochondrial membrane is lost during freezing as we already know, the complexes of the electron transport chain are still intact and functional. Therefore, providing substrates that directly feed the ETC complexes should be enough to elicit electron flux and consume oxygen. Importantly, this approach enables the analysis of maximal uncoupled mitochondrial respiration in previously frozen samples such as patient-derived tissue from biobanks. The underlying idea is logic and simple. Eventually, analysis of (functional) supercomplexes can be carried out both in fresh and frozen samples with minimal problems, and the same should apply to oxygen consumption. Accordingly, while reading this manuscript I was asking why I didn't think of this idea myself. Overall, data are well performed and discussed, and most of the necessary controls are already in place. While data on OCR are strong, the section on mitochondrial quantification using a fluorescent indicator is rather weak and does not add so much to the overall story. Some points need more attention:

- I do not understand the difference in basal OCR in some panels, including 1D, 1G, 3C and 3E. Basal respiration should correspond to state 1 (no substrates, no ADP) and no difference between fresh and frozen samples should be present. Indeed, according to the figures and legends, both substrates and ADP are injected using the first port of seahorse reader. Therefore, traces should begin (i.e. first time point) with comparable OCR. I think this is simply due to erroneous labeling of the figures. Most likely, substrates (pyr/mal in panel 1A, succ/rot in panel 1D and NADH in panel 1G) are already present at the beginning of the experiment (state 4 is thus measured here), and only ADP is added with the first injection. Please fix this problem and provide control experiments showing state 1 respiration.*

• As far as I know, Chance and Williams define state 2 respiration as a condition with high ADP and low substrates. Conversely, figure legend 1 reports "state 2 (substrate without ADP)", that should be instead considered as state 4. Accordingly, respiratory control ratios (RCRs) are defined as state3/state4 (not state3/state2 as shown).

The reviewer is right on the nomenclature by Britton and Chance. The confusion is that we used the nomenclature used by Nichols and Ferguson in Bioenergetics book, as it was more intuitive in which state 1 is mitochondria alone, state 2 with substrates, state 3 with ADP and state 4 ADP exhaustion or 4o adding oligomycin. We will correct the nomenclature according to Britton and Chance to avoid confusions. Thus, the assumption of the reviewer on the cause for the differences in initial OCR is correct.

• As an additional control, the authors should test the effect of atractyloside. It should inhibit state 3 respiration in fresh but not in frozen samples.

The reviewer is also right in his/her assumptions on how the samples were placed in the seahorse plate: samples are in the presence of substrates (no ADP) in the well and ADP is injected in port A for all the conditions. We have now corrected the labeling in the different panels of Figure 1. In addition, we have performed an experiment comparing fresh versus frozen using liver mitochondria and liver homogenates. We have started the assay with samples in the absence of substrates (state 1); then injected substrates in port A (state 4); substrates plus ADP in port B (state 3); atractyloside (ANT inhibitor) in port C and azide in port D. The results are presented in the figure below and in panels J and K on the new Figure 1.

• The least convincing part of this work is the use of Mitotracker DeepRed to normalize mitochondrial content. I see several problems here.

1) Figure 6A demonstrates that some parameters can impact on MTDR fluorescence, since Ca²⁺ treatment causes an approximately 25% decrease of fluorescence with no apparent explanation.

The calcium concentration used in this assay is in the high range (2mM) so presumably, mitochondrial swelling and membrane rupture occurs. Note that for mitochondrial swelling assays to measure the permeability transition pore, we normally use 0.4mM. We have now repeated the experiment with lower calcium concentrations and showed that MTDR fluorescence is not affected. We have now substituted the panel A in Figure 6 for a new panel including the new data

2) Figure 6B clearly shows that the relationship between protein content and MTDR fluorescent is linear only over a narrow range (up to 10 ug for isolated mitochondria, without considering that the linearity in the case of total homogenate looks questionable). In a real-life situation, it would be difficult to know if MTDR signal is at saturation or not only looking at raw fluorescence.

We agree with the reviewer that it might difficult to predict in real-life situations how much sample needs to be added in order to not reach the saturation limit in the assay. We have tested different protein amounts depending on the sample source and found that, if we use the same protein amount in micrograms that we use for the Seahorse assays, we are within the linear range. Every fluorescence and colorimetric assay has the challenge of using sample amounts that read in the linear range of the assay. As additional controls, a standard curve using MTDR as well as dilutions of the original sample can be used to measure mitochondrial content. These results are now part of the new Figure S5 (panels C and D).

3) When comparing figure 7F to 7H, MTDR vs protein show very different results. According to MTDR signal, mitochondrial content is twice higher in the POLg sample, i.e. an effect that should be well visible with other techniques. According to Western blot, the increase in POLg is very limited, thus questioning results obtained with MTDR. Overall, I think the main problem here is that we don't know exactly how MTDR works, and it is thus difficult to understand its strengths and limitation. I think this part can be omitted with no impact on the overall conclusions.

We agree with the reviewer that we do not exactly understand the mechanisms by which MTDR stains mitochondria. However, we still think that the information that it provides is very useful when assessing mitochondrial content.

There is not a single method for measuring mitochondrial mass that is encompassing the fact that the various components of mitochondria are expanding equally. One has to select a surrogate marker for mitochondrial mass and each selection brings with it a potential error that can be addressed in a thorough analysis of the various components of mitochondria. Among the most commonly used are mtDNA/nDNA ratio, western blot analysis and citrate synthase (CS) activity. However, the one, which is suitable for relatively higher throughput, is a fluorescent-based analysis by either high throughput imaging or plate reader. Keeping in mind that mitochondrial mass measurement is not the subject of this publication, our approach was to use the 3 most approachable methods for mitochondrial mass measurements and describe their limitations. We include data with CS activity, WB and fluorescence analysis of MTDR in a plate reader. We believe that MTDR offers an alternative that many readers will consider and thus it is important to examine it and outline its limitations. To compare the methods, we have included in the revised version a comparison of data obtained from CS vs MTDR. The comparison demonstrate that at least in the range of 2-30 μ g of lysate, the 2 approaches are linearly correlated

In some models of mitochondrial pathology, the cell tries to compensate for the energy deficiency through increased mitochondrial biogenesis (Garone and Viscomi, 2018; Sanchis-Gomar et al., 2014; Uittenbogaard and Chiaramello, 2014; Wredenberg et al., 2002). When bioenergetics analysis is performed in isolated mitochondria, the activity per protein and per mitochondrial content (MTDR) follows the same trend, as expected. However, when the mitochondrial activity is measured in homogenates, the lysate also contains other organelles and the cytosol in addition to mitochondria, all contributing to protein levels. Nevertheless, some lysates might contain more mitochondria than others. By including the measurement of mitochondrial mass, we can refer the mitochondrial activity in the lysates as activity per cell (per protein) or per mitochondria (per MTDR). Under some pathological conditions, increasing mitochondrial mass allows the cell to compensate for bioenergetic deficiencies by having more, albeit dysfunctional, mitochondria. To that point, measuring mitochondrial content/mass is very important. In the POLgMT model, mtDNA accumulates mutations with age that can affect mitochondrially encoded protein turnover without affecting nuclearly encoded protein turn over or levels. Moreover, steady state levels of

some proteins do not reflect the abundance of mitochondria or the assembled complexes since they still can be detected in rho0 cells lacking mtDNA.

Referee #3:

This paper describes measurements of mitochondrial respiratory chain activity from various tissues and biological sources. The aim is purely methodological: to demonstrate that activity is maintained despite freezing and thawing the samples, which obviously saves a lot of material. My major concern is that there is no actual conceptual novelty in this work; its value may be considerable but it is purely methodological. This is more a matter of decision by the Journal's Editors. To my mind this work is not suitable for the EMBO Journal but should be published in a methodological forum.

We thank the referee for the comment. This manuscript is submitted to EMBO J Resources format, which is a forum in which EMBO is inviting methodological papers.

I have somewhat minor concerns related to the method of measuring oxygen consumption. There is no information of how the actual numbers of O₂ consumption were obtained. What was the temperature (oxygen solubility is highly temperature dependent)? How was the oxygen concentration determined? How did the authors control for oxygen diffusion into the microplate wells? How was the zero oxygen point established?

The method developed in this study is to be used with any platform of oxygen consumption measurement, such as Seahorse XF, Oroboros, Oxygraph and others. We did not get into the details of how the instrument is working as we did not make any changes to the instrument used for oxygen tension measurements. To demonstrate the development described here, we are using the seahorse platform for oxygen consumption measurements. In their website, publications and patents, Seahorse (Agilent) is providing information about the way the measurements are obtained and how the XF instrument takes into account the temperature (37C) and the oxygen solubility at that temperature to convert the mV into nmol of oxygen that we are using as a readout.

2nd Editorial Decision

1st March 2020

Thank you for submitting a revised version of your manuscript. It has now been seen by two of the original referees, whose comments are shown below.

As you will see, they both find that all criticisms have been sufficiently addressed and recommend the manuscript for publication. However, before we can officially accept the manuscript, there are a few editorial issues concerning text and figures that I need you to address.

REFeree REPORTS

Referee #1:

Clearly the authors made a significant effort to address all concerns of the reviewers. The revision is satisfactory and I have no further comments.

Referee #2:

The authors successfully addressed my original concerns. I think the paper is now improved and ready to be published

2nd Revision - authors' response

11th March 2020

The Authors have made the requested editorial changes.

Corresponding Author Name: Orian S Shirihai, Rebeca Acin-Perez and Linsey Stiles

Journal Submitted to: EMBO J

Manuscript Number: EMBOJ-2019-104073R